# Co-targeting the tumor endothelium and P-selectin-expressing glioblastoma cells leads to a remarkable therapeutic outcome

Shiran Ferber[1†], Galia Tiram[1†], Ana Sousa-Herves[2†], Anat Eldar-Boock[1], Adva Krivitsky[1], Anna Scomparin[1], Eilam Yeini[1], Paula Ofek[1], Dikla Ben-Shushan[1], Laura Isabel Vossen[2], Kai Licha[2], Rachel Grossman[3], Zvi Ram[3], Jack Henkin[4], Eytan Ruppin[1,5,6,7], Noam Auslander[5,7], Rainer Haag[2], Marcelo Calderón[2], Ronit Satchi-Fainaro[1,8*]

[1]Department of Physiology and Pharmacology, Sackler Faculty of Medicine, Tel Aviv University, Tel Aviv, Israel; [2]Institute of Chemistry and Biochemistry, Freie Universität Berlin, Berlin, Germany; [3]Department of Neurosurgery, Tel Aviv Sourasky Medical Center, Tel Aviv, Israel; [4]Chemistry of Life Processes Institute, Northwestern University, Evanston, United States; [5]Center for Bioinformatics and Computational Biology, University of Maryland, College Park, United States; [6]Blavatnik School of Computer Sciences, Tel Aviv University, Tel Aviv, Israel; [7]Department of Computer Science, University of Maryland, College Park, United States; [8]Sagol School of Neurosciences, Tel Aviv University, Tel Aviv, Israel

*For correspondence:
ronitsf@post.tau.ac.il

[†]These authors contributed equally to this work

**Competing interests:** The authors declare that no competing interests exist.

**Abstract** Glioblastoma is a highly aggressive brain tumor. Current standard-of-care results in a marginal therapeutic outcome, partly due to acquirement of resistance and insufficient blood-brain barrier (BBB) penetration of chemotherapeutics. To circumvent these limitations, we conjugated the chemotherapy paclitaxel (PTX) to a dendritic polyglycerol sulfate (dPGS) nanocarrier. dPGS is able to cross the BBB, bind to P/L-selectins and accumulate selectively in intracranial tumors. We show that dPGS has dual targeting properties, as we found that P-selectin is not only expressed on tumor endothelium but also on glioblastoma cells. We delivered dPGS-PTX in combination with a peptidomimetic of the anti-angiogenic protein thrombospondin-1 (TSP-1 PM). This combination resulted in a remarkable synergistic anticancer effect on intracranial human and murine glioblastoma via induction of Fas and Fas-L, with no side effects compared to free PTX or temozolomide. This study shows that our unique therapeutic approach offers a viable alternative for the treatment of glioblastoma.
DOI: https://doi.org/10.7554/eLife.25281.001

## Introduction

Glioblastoma is the most common form of primary brain tumor and is one of the most aggressive forms of cancer (*Wen and Kesari, 2008*). Without treatment, the median survival is approximately 3 months (*Schapira, 2007*). The current standard of treatment involves maximal surgical resection followed by concurrent radiation therapy and chemotherapy with the DNA alkylating agent temozolomide (TMZ) (*Stupp et al., 2005*). With this regimen, the median survival is increased to approximately 14 months (*Stupp et al., 2005*). Glioblastoma's diffusive and invasive nature makes complete removal of the tumor by conventional surgery nearly impossible. Therefore, tumor

recurrence is inevitable and only 1.8% of patients survive more than three years (*Scott et al., 1998*; *Wen and Kesari, 2008*). Management options for recurrent glioblastoma include a second cycle of surgery, radiation and/or chemotherapy, for which the majority of treatment options remain palliative. In some patients, surgery is accompanied by an implantation of biodegradable wafers carrying the chemotherapeutic agent carmustine (GLIADEL) (*Brem and Gabikian, 2001*). Various other treatments using cytotoxic chemotherapy have shown disappointing results with an average overall survival (OS) of 3–9 months (*Lamborn et al., 2008*; *Yung et al., 2000*). Additionally, bevacizumab, an anti-vascular endothelial growth factor (VEGF) monoclonal antibody, was FDA-approved for the treatment of recurrent glioblastoma in 2009 (*Kreisl et al., 2009*). Although progression-free survival was slightly prolonged and an overall lowering of glucocorticoid requirements to control tumor-related edema was shown in patients with recurrent glioblastoma (*Friedman et al., 2009*), treatment with bevacizumab among patients with newly diagnosed glioblastoma did not exhibit improvement in OS of patients (*Chinot et al., 2014*; *Gilbert et al., 2014*). As most patients develop resistance to current therapy, poor patient outcomes with minimally effective treatment options necessitate the development of novel agents that target relevant therapeutic pathways.

The high prevalence of resistance to conventional therapy in glioblastoma (30–60% of patients) (*Weller et al., 2010*) and poor outcome have led to several studies screening for new potential drugs that typically target other vital cellular functions. Mitotic inhibitors, such as paclitaxel (PTX), were found to be among the most potent drugs (*Jiang et al., 2014*). This class of drug binds to polymerized tubulin and inhibits the dissociation rate, leading to M phase cell cycle arrest (*Schiff et al., 1979*). PTX was shown to be highly potent against glioblastoma cells in vitro (*Cahan et al., 1994*), but systemic delivery was associated with poor pharmacokinetics and neurotoxicity (*Berger et al., 1997*). Furthermore, hypersensitivity was an additional toxicity which was not attributed to the drug itself, but rather to its polyoxyethylated castor oil solubilizing vehicle, Cremophor EL (CrEL) (*Mielke et al., 2005*). Clinically significant neurotoxicity appears after administration of a cumulative dose of around 1,500 mg/m$^2$ (*Briasoulis et al., 2002*). This suggests that a smart delivery system, that will facilitate its selective transport to the cancerous tissue, might reduce or even eliminate these adverse effects and improve its pharmacokinetics and tumor accumulation.

A well-designed polymeric drug delivery system (DDS) improves the therapeutic index of anticancer agents. An ideal DDS can increase the half-life of low-molecular-weight drugs, improve water-solubility (*i.e.,* CrEL-free formulations) and cause selective tumor tissue accumulation, all while enhancing anticancer efficacy, reducing toxicity and limiting drug resistance (*Markovsky et al., 2012*). These delivery systems exploit the natural (blood circulation and extravasation-dependent targeting – 'passive') distribution pattern of a drug-carrier in vivo, which is based upon the enhanced permeability and retention (EPR) effect (*Matsumura and Maeda, 1986*). Despite the fact that excessive angiogenesis and disruption of the blood-brain barrier (BBB) are hallmarks of glioblastoma, restricted permeability of the BBB to systemic therapy is still a persistent challenge. Previous studies have shown that blood circulation and extravasation-dependent targeting of DDS to brain tumors does not lead to sufficient drug penetration to the tumor, and thus results in very little improvement in therapeutic efficacy (*Muldoon et al., 2007*). The drawbacks of passive targeting have led to the investigation of active ligand directed-targeting (*Kang et al., 2015*) or local delivery to the resected tumor cavity (*Brem and Gabikian, 2001*; *Ranganath et al., 2009*) for PTX delivery. Most targeted DDS that are currently in preclinical or phase I trials are designed to target BBB-related moieties (*e. g.* systems conjugated to ligands or antibodies to transferrin receptor, insulin, low-density lipoprotein (LDL) and LDL receptor-related protein-1 and −2) (*Pardridge, 2010*). While crossing the BBB is of significant importance, drug efficacy in brain tumors requires diffusion in the brain extracellular space (ECS) in addition to the ability to cross the cancer cells membrane for internalization. Diffusion across the ECS in the brain is greatly limited by high cerebrospinal fluid turnover rate and efflux system, which results in drug elimination (*Huynh et al., 2006*). Even with technological advances in surgical techniques, malignant glioma often recur within 1–2 cm of the original tumor site (*Hochberg and Pruitt, 1980*). Therefore, local delivery of drugs to the resected tumor cavity has emerged as a promising therapeutic alternative to treat residual disease (*Hou et al., 2006*). Unfortunately, diffusion of drugs is restricted to up to 3 mm from the injection site in case of local delivery (*Fleming and Saltzman, 2002*). Therefore, we hypothesized that in order to achieve efficient targeting that will result in significant therapeutic efficacy, a DDS should be targeted to both the tumor microenvironment and the cancer cells, preferably internalizing via receptor-mediated endocytosis.

In this study, we demonstrate the delivery of PTX with dendritic polyglycerol sulfate (dPGS), targeting glioblastoma microenvironment as well as the cancer cells themselves. We have previously shown that dPGS nanocarriers, depending on size and degree of sulfation of the polymer core, bind and inhibit both L-selectin and P-selectin (*Dernedde et al., 2010*; *Khandare et al., 2010*; *Sousa-Herves et al., 2015*; *Weinhart et al., 2011a*, *Weinhart et al., 2011b*). Selectins are carbohydrate-binding proteins that are physiologically expressed on the surface of endothelial cells (P-selectin and E-selectin), activated platelets (P-selectin) and leukocytes (L-selectin) (*Läubli and Borsig, 2010*). Selectins are also involved in tumor cell binding and promote their invasion by supporting a permissive metastatic microenvironment and protecting them from recognition by immune cells (*Preusser et al., 2012*). To target glioblastoma, we have chosen dPGS with the highest affinity to selectins, which consists of 90% sulfate groups and a 5 KDa molecular weight core (*Dernedde et al., 2010*). Sulfate groups mimic the naturally occurring ligand of P/L-selectin and, as opposed to endogenous ligands, significantly increase binding affinity owing to their high number of functional groups. Here we show that dPGS can target not only the tumor microenvironment, but also the cancer cells, via binding to P-selectin expressed on glioblastoma cells. To the best of our knowledge, expression of P-selectin by glioblastoma cells was not previously reported. Utilizing a dendritic polyglycerol (dPG)-based structure as the DDS allows us to take advantage of its unique architecture. dPG forms three-dimensional hyperbranched nanostructures that have been shown before to efficiently extravasate through the compromised BBB (*Ofek et al., 2016*). They are also water-soluble, easily tuned in size and functional groups, and synthesized on a kilogram scale (*Calderón et al., 2014*; *Calderón et al., 2011*; *Hussain et al., 2013*; *Weinhart et al., 2011b*).

The benefits of combination therapy for cancer have been widely reported and reviewed (*Eldar-Boock et al., 2013*). We postulated that combining PTX with another therapeutic agent can potentially result in a synergistic anti-tumor effect while using a moderate PTX dosing regimen, thereby reducing its dose-limiting toxicities. Given the highly angiogenic nature of glioblastoma, our approach was to combine dPGS-PTX with a potent anti-angiogenic agent. Thrombospondin-1 (TSP-1) is a key endogenous angiogenesis inhibitor, and its expression is often lost or decreased throughout malignant transformation (*Sargiannidou et al., 2001*). In gliomas, TSP-1 downregulation results in accelerated disease (*Hsu et al., 1996*) and loss of TSP-1 correlates with transition from low-grade astrocytoma to glioblastoma (*Almog et al., 2009*). The interactions of TSP-1 with multiple cell surface proteins determine its diverse functions. The anti-angiogenic activity of TSP-1 is predominantly attributed to type one thrombospondin repeats (TSRs) that interact with a fatty acid translocase receptor on the endothelial cell membrane, CD36 (*Simantov and Silverstein, 2003*). TSRs mimetic peptides have been successfully used to block angiogenesis and tumor growth in preclinical models (*Campbell et al., 2011*; *Rusk et al., 2006*). ABT-510 (Abbott Laboratories) is a nonapeptide with a single D-isoleucine replacement that resulted in a 1,000-fold increase of its anti-angiogenic activity (*Haviv et al., 2005*). ABT-510-dependent apoptosis of tumor endothelial cells was shown to be induced by their upregulation of Fas ligand, and was amplified by low doses of chemotherapeutic agents since the latter increased endothelial cells presentation of CD95 death receptor, an effect seen with docetaxel (DTX) above 1 nM (*Yap et al., 2005*). The peptide was also shown to normalize tumor vasculature, and thereby enhance the delivery and efficacy of both cisplatin and PTX in a syngeneic model of ovarian cancer (*Campbell et al., 2010*). Preclinically active as monotherapy, it was evaluated in phase I and II clinical trials. ABT-510 mitigated orthotopic glioma growth in a mouse model by increasing apoptosis in tumor microvascular endothelial cells (*Campbell et al., 2010*). In a phase I clinical trial for the treatment of newly-diagnosed glioblastoma in combination with the standard of care (*i.e.*, TMZ and radiation), ABT-510 exhibited a favorable safety profile at a dose of up to 200 mg daily for 6 days and additional 4 weeks as monotherapy (*Nabors et al., 2010*). In this study, improved clinical outcome was associated with downregulation of the pro-angiogenic factors fibroblasts growth factor-1 (FGF-1) and tyrosine kinase with immunoglobulin-like and epidermal growth factor (EGF)-like domains-1 (TIE1). However, another phase I clinical trial showed a poor pharmacokinetic profile of the peptide, demonstrating only a one hour clearance half-life (*Hoekstra et al., 2005*). The short half-life of ABT-510 and the difficulty to achieve effective plasma concentrations may explain failures in phase II clinical trials both as a monotherapy and in combination with chemotherapy (*Ebbinghaus et al., 2007*). A second-generation octapeptide was developed (ABT-898) with improved potency and slower clearance (*Henkin and Volpert, 2011*). Thus far, ABT-898 has been evaluated in preclinical models of epithelial ovarian cancer and of uveal melanoma in mice

(*Campbell et al., 2011*; *Wang et al., 2012*) and in treatment of spontaneously-occurring soft tissue sarcoma in dogs (*Sahora et al., 2012*). It was efficacious in all these settings. Here, we show that ABT-898 has a therapeutic potential for glioblastoma as well, by stabilizing blood vessels and reducing the angiogenic potential of established tumors.

In this study, we demonstrate that combination therapy of PTX and TSP-1 peptidomimetic (TSP-1 PM), ABT-898, in conjunction with improved selective targeting to the desired cancer tissue by dPGS, remarkably prolongs OS of mice bearing human and murine orthotopic glioblastoma. TSP-1 PM was delivered at a high dose of 100 mg/kg. PTX was delivered at a reduced accumulated dose of 90 mg/kg. This correlates to 270 mg/m$^2$, which is considerably lower than the maximal tolerated cumulative dose reported (*Briasoulis et al., 2002*). This combination therapy was well tolerated by treated mice, with no neurological or systemic toxicities demonstrated. This approach represents a potential therapeutic alternative for glioblastoma patients who fail to respond to the standard of care.

## Results

### Synthesis and characterization of dendritic PTX conjugates

dPGS was selected as the dendritic platform for conjugation to PTX due to its non-toxicity, both in vitro and in vivo (*Dernedde et al., 2010*; *Khandare et al., 2010*), and its exceptionally high binding affinity to selectins involved in the inflammatory process, such as L- and P- selectins. In addition, dPGS can be prepared on a multigram scale and with different sulfation degrees, architectures, and molecular weights (*Weinhart et al., 2011b*).

The binding affinity of dPGS towards selectins is dependent on the molecular weight and degree of sulfation (*Reichert et al., 2011*; *Weinhart et al., 2011a*). Thus, among the different dPGS reported by our group, we have selected one with a molecular weight of around 11 kDa, 90% degree of sulfation, and 5% amine groups because it displayed strong binding affinities for cellular targets, and hence seemed to be a good candidate for the design of a self-targeting macromolecular conjugate (*Dernedde et al., 2010*; *Weinhart et al., 2011a*; *Weinhart et al., 2011b*). As non-targeting control, a non-sulfated analogue of similar molecular weight and number of amine groups (dPG amine, $M_n$ 10 kDa, 10% NH$_2$) has been selected (*Figure 1—figure supplement 1*).

We have recently reported the synthesis of dPGS conjugated to PTX through an acid cleavable ester bond (*Sousa-Herves et al., 2015*). However, in that case, premature drug release was observed due to hydrolysis of the ester linkages by the action of hydrolytic enzymes before the conjugate could reach acidic intracellular compartments. With the aim of overcoming such limitation while maintaining controlled drug release upon cellular internalization, we have employed a pH-cleavable hydrazone bond for conjugation of PTX to the dendritic platforms.

In order to prepare the conjugates, a (6-maleimidocaproyl) hydrazine (EMCH) linker (*Calderón et al., 2011*; *Kratz, 2007*) was first introduced into the C-2'-OH-position of PTX. Then, the NH$_2$ groups of dPGS amine or dPG amine were reacted with 2-iminothiolane, followed by a selective Michael addition between the maleimide group of the PTX-EMCH and the sulfhydryl groups from thiolated dPGS or dPG (*Figure 1A*).

The chemical structure of dPGS-PTX is depicted in *Figure 1B*. The resulting sulfated targeted conjugate dPGS-PTX and control dPG-PTX had approximately 1 mol of PTX per mol of conjugate, as estimated by $^1$H NMR (*Table 1*, *Figure 1—figure supplement 2*). In addition, the size and zeta potentials of the conjugates were further determined by dynamic light scattering (DLS) at physiological pH (*Figure 1C*; *Table 1*). dPGS-PTX had a hydrodynamic diameter of about 46 nm which is ideal for exploiting the EPR effect. dPG-PTX was smaller, with a hydrodynamic diameter of about 4 nm. These size differences might be attributed to a supramolecular arrangement of dPGS, resulting from the high affinity of the negatively-charged sulfate groups and positively-charged amine groups, as opposed to the dPG structure that contains neutral hydroxyl groups instead of sulfates (*Figure 1—figure supplement 1B*). The size of dPGS-PTX was further validated by scanning electron microscopy (SEM) (*Figure 1D*). Similarly to the size obtained by the DLS, dPGS-PTX was shown to have a diameter of 50 nm. SEM imaging of dPG-PTX particles was impossible, due to their small size compared with the instrument's resolution relating to our organic nanoparticles. At physiological pH, the amine surface functional groups are expected to be partially protonated, while all the sulfate groups

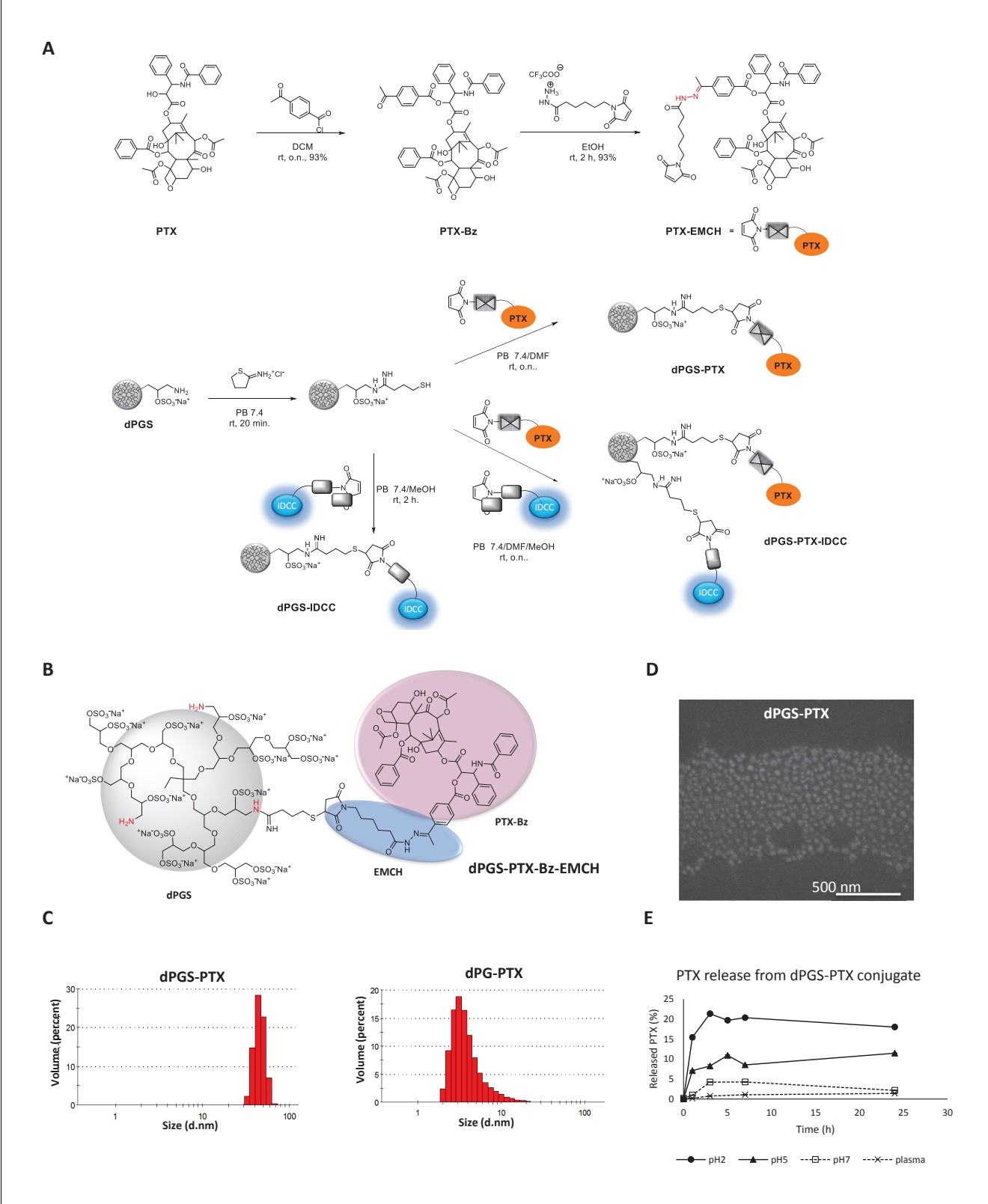

**Figure 1.** Synthesis and physicochemical characterization of dPG-PTX nanoconjugates. (**A**) Scheme depicting the synthesis of PTX-EMCH and dPGS conjugates. (**B**) Chemical structure of dPGS-PTX. (**C**) Size distribution of dPGS-PTX and dPG-PTX nanoconjugates, obtained by zetasizer ZS. Data is representative of 3 individual experiments. (**D**) SEM image of dPGS-PTX nanoconjugate as obtained by Quanta 200 FEG Environmental SEM. (**E**)
*Figure 1 continued on next page*

*Figure 1 continued*

Analysis of PTX cumulative release (%) from dPGS-PTX at different pH values and in human plasma, as obtained from HPLC measurements. Data represent mean ± s.d. of three independent experiments.

DOI: https://doi.org/10.7554/eLife.25281.002

The following figure supplements are available for figure 1:

**Figure supplement 1.** Idealized chemical structure of dendritic nanocarriers based on polyglycerol.

DOI: https://doi.org/10.7554/eLife.25281.003

**Figure supplement 2.** $^1$H NMR (700 MHz, DMSO-d$_6$) of dPGS-PTX and dPG-PTX nanoconjugates.

DOI: https://doi.org/10.7554/eLife.25281.004

will be negatively-charged. As expected, zeta potential measurements for dPGS-PTX, containing 90% surface sulfate groups and 5% amine groups (*Figure 1—figure supplement 1A*), showed a substantial negative charge of about −47 mV, while dPG-PTX, containing 10% amine surface groups and 90% of the mostly non-ionized hydroxyl groups, had a zeta potential of + 16.5 mV, similar to previous reports of dPG bearing the same percentage of cationic groups (*Khandare et al., 2010*).

In order to demonstrate the stability of the hydrazone bonds at physiological pH and in the presence of esterases and other hydrolytic enzymes present in plasma, the release profile of PTX from dPGS-PTX conjugate was analyzed by HPLC. As expected, a continuous PTX release was observed at pH two and pH 5, while marginal PTX release was detected at pH seven or in plasma after 24 hr (*Figure 1E*). The peripheral amine groups displayed on these dendritic structures allowed the simultaneous incorporation of PTX and indodicarbocyanine (IDCC) dyes, which was exploited to monitor the in vitro and in vivo behavior of the conjugates. Such multifunctional conjugates were synthesized in a sequential one pot approach (*Figure 1A*) (*Baabur-Cohen et al., 2017*; *Calderón et al., 2011*). dPGS and dPG were first reacted with 2-iminothiolane, followed by reaction with PTX-EMCH. Subsequently, in a second step, a solution of 2S-IDCC-maleimide (*Gröger et al., 2013*; *Licha et al., 2001*) was added to achieve dye conjugation (*Figure 1—figure supplement 1C*). After purification, the formation of the multifunctional conjugates dPGS-PTX-IDCC and dPG-PTX-IDCC was confirmed by HPLC and UV-Vis spectroscopy. Finally, following the same synthetic strategy, conjugates containing the IDCC dyes but lacking PTX were prepared for comparison.

## Cellular uptake and intracellular trafficking

In order to evaluate the cellular uptake of the dendritic conjugates into glioblastoma cells, patient-derived hGB1 cells were incubated with IDCC-labeled dPGS-PTX and dPG-PTX for different periods of time. Flow cytometry analysis evaluating the uptake of IDCC-labeled dendritic conjugates demonstrated that they exhibit different internalization kinetics. dPGS-PTX-IDCC rapidly internalized into the cells within minutes, whereas the non-targeted conjugate, dPG-PTX-IDCC, internalized after more than 4 hr (*Figure 2A*). To evaluate whether the internalization of dPGS-PTX is P-selectin-dependent, we used a low molecular weight P-selectin inhibitor (KF 38789, Tocris). This compound has been previously shown to inhibit the binding of U937 lymphocytes to immobilized P-selectin immunoglobulin, with an IC$_{50}$ value of 1.97 μM (*Ohta et al., 2001*). Treatment with the P-selectin inhibitor prior to treatment with the dendritic conjugates inhibited dPGS-PTX-IDCC uptake in a dose-dependent manner, while not affecting the uptake of dPG-PTX-IDCC (*Figure 2B*). This demonstrated that dPGS-PTX internalizes into the cells via P-selectin. To further show the advantage of dPGS-PTX compared to its non-targeted control, we treated patient-derived glioblastoma tumor spheroids with IDCC-labeled dendritic conjugates. Similar to the results received in 2D culture,

**Table 1.** Summary of physicochemical characterization of dendritic conjugates

| Compound | Molar ratio PTX/Conjugate | Molecular weight (Da) | Hydro-dynamic diameter (nm) | PDI | Diameter by SEM (nm) | Zeta potential (mV) |
|---|---|---|---|---|---|---|
| dPGS-PTX | ~1 | 12,807 | 45.9 ± 6.7 | 0.693 | 50 ± 20 | −47.1 ± 5.1 |
| dPG-PTX | ~1 | 11,207 | 4.2 ± 2.6 | 0.268 | NA | 16.5 ± 4.3 |

DOI: https://doi.org/10.7554/eLife.25281.005

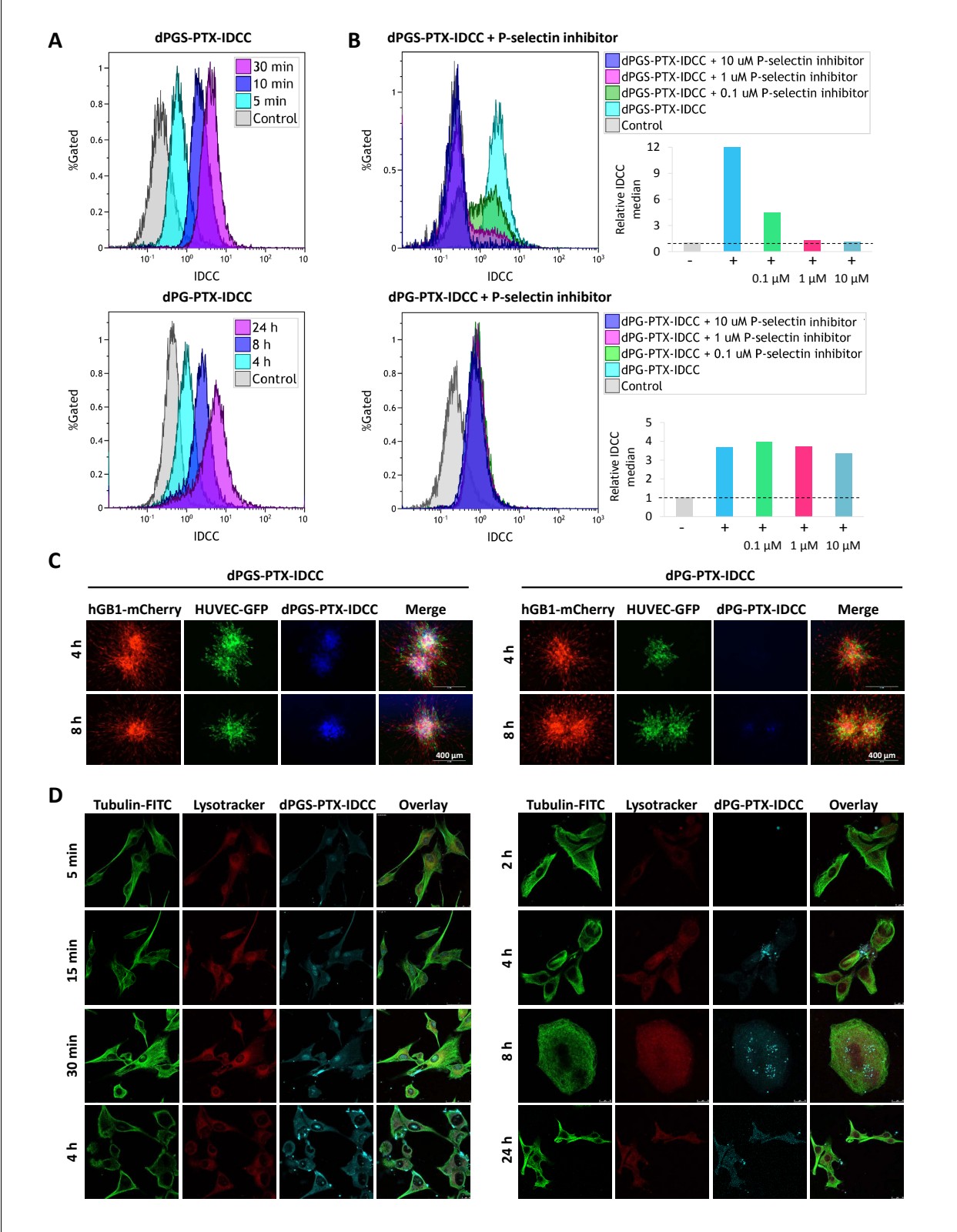

**Figure 2.** Targeted sulfated conjugate dPGS-PTX-IDCC efficiently internalizes into human glioblastoma cells and spheroids via P-selectin. (**A**) Flow cytometry analysis of the cellular uptake of dPGS-PTX-IDCC following 5, 10 and 30 min incubation and dPG-PTX-IDCC following 4, 8 and 24 hr incubation in patient-derived glioblastoma cells (hGB1). (**B**) Flow cytometry analysis of the cellular uptake of dPGS-PTX-IDCC and dPG-PTX-IDCC following treatment with 0.1, 1 and 10 μM P-selectin inhibitor. (**C**) Internalization of IDCC-labeled dendritic conjugates (blue) into tumor spheroids

*Figure 2 continued on next page*

*Figure 2 continued*

composed of mCherry-labeled hGB1 cells (red), GFP-labeled HUVEC (green) and astrocytes. (D) Cellular uptake and intracellular trafficking of dPGS-PTX-IDCC and dPG-PTX-IDCC conjugates in U-87 MG cells. The figure depicts representative confocal images of IDCC-labeled dendritic conjugates (cyan), tubulin (green) and the lysosome (red).

DOI: https://doi.org/10.7554/eLife.25281.006

The following figure supplement is available for figure 2:

**Figure supplement 1.** Targeted sulfated conjugate dPGS-PTX colocalizes with tubulin efficiently and rapidly.

DOI: https://doi.org/10.7554/eLife.25281.007

dPGS-PTX internalized into the spheroids more rapidly and efficiently compared to dPG-PTX (*Figure 2C*).

The pH sensitive EMCH linker between PTX-Bz and the dendritic conjugates is cleaved at acidic pH typical for the lysosome, thus releasing the active drug into the cytoplasm. PTX can then bind and stabilize the tubulin, arresting cells in mitosis. To investigate the intracellular trafficking mechanisms of the dendritic conjugates, cells treated with IDCC-labeled dPGS-PTX and dPG-PTX were stained for tubulin and their lysosomes were labeled using lysotracker. As demonstrated previously by flow cytometry, following 5 min incubation, the targeted sulfated dendritic conjugate was already detected at the cells cytoplasm and gradually accumulated for 4 hr. Tubulin damage was observed with time (*Figure 2D*). Co-localization analysis between dPGS-PTX-IDCC and tubulin supported these results. Increased localization of the targeted dendritic conjugate in the lysosome at the first 30 min, followed by reduction at the 2 hr time point, indicated that PTX was released into the cytoplasm (*Figure 2—figure supplement 1A*). In contrast, the non-targeted dendritic conjugate (dPG-PTX-IDCC), was observed inside the cells only after 4 hr and gradually accumulated following 8 hr. During that time period, the tubulin seemed intact. Tubulin damage was observed only after 24 hr incubation (*Figure 2D*). Co-localization analysis between dPG-PTX-IDCC and tubulin showed an increase in co-localization following 4 hr and 8 hr incubation, but to a lesser extent compared to the targeted conjugate (*Figure 2—figure supplement 1B*). These results are in accordance with previous studies demonstrating efficient internalization of sulfated dPG conjugates into several cancer cell lines (*Khandare et al., 2010*; *Sousa-Herves et al., 2015*).

## P-selectin is highly expressed in glioblastoma

Since P- and E-Selectins are typically expressed on the surface of activated platelets and endothelial cells, differences between the sulfated targeted and non-targeted conjugates were only expected to be observed in vivo. The rapid internalization of dPGS-PTX (*i.e.*, 5 min) suggested an internalization mechanism other than fluid-phase pinocytosis via simple endocytosis. Therefore, P-Selectin expression was evaluated in U-87 MG cells. As shown in *Figure 3A*, U-87 MG expressed high P-Selectin levels on their surface. In order to evaluate whether P-Selectin overexpression is a single cell line phenomenon, fresh glioblastoma tissues were collected and two new patient-derived glioblastoma cell lines were isolated. These two new cells lines and an additional murine glioblastoma cell line, GL261, were evaluated for their P-selectin expression by FACS. Surprisingly, P-Selectin overexpression was observed in all tested human and murine samples (*Figure 3A*). Furthermore, in all cell lines tested, the shift in fluorescence signal observed by FACS analysis of a single peak indicates a single population expressing P-Selectin, rather than overexpression of a sub-population. Thus, suggesting that these major differences in internalization between the sulfated and non-sulfated conjugate might be a result of receptor-mediated endocytosis by dPGS-PTX. We therefore postulated that dPGS-PTX would not only target the tumor microenvironment, by binding P/E-Selectin on endothelial cells and activated platelets, but also the cancer cells population, thus resulting in improved selective targeting, enhanced cellular internalization by receptor-mediated endocytosis and better therapeutic outcome. To validate the therapeutic potential of our targeted nanoconjugate in vivo, we evaluated P-selectin expression in xenogeneic and syngeneic mouse models of human and murine glioblastoma. P-Selectin was highly expressed in human U-87 MG and murine GL261 tumors, whereas almost no positive staining was observed in the adjacent normal brain (*Figure 3B*). P-selectin was also found to be abundantly expressed in glioblastoma patient specimens (*Figure 3B*), suggesting a clinical relevance of our therapeutic approach.

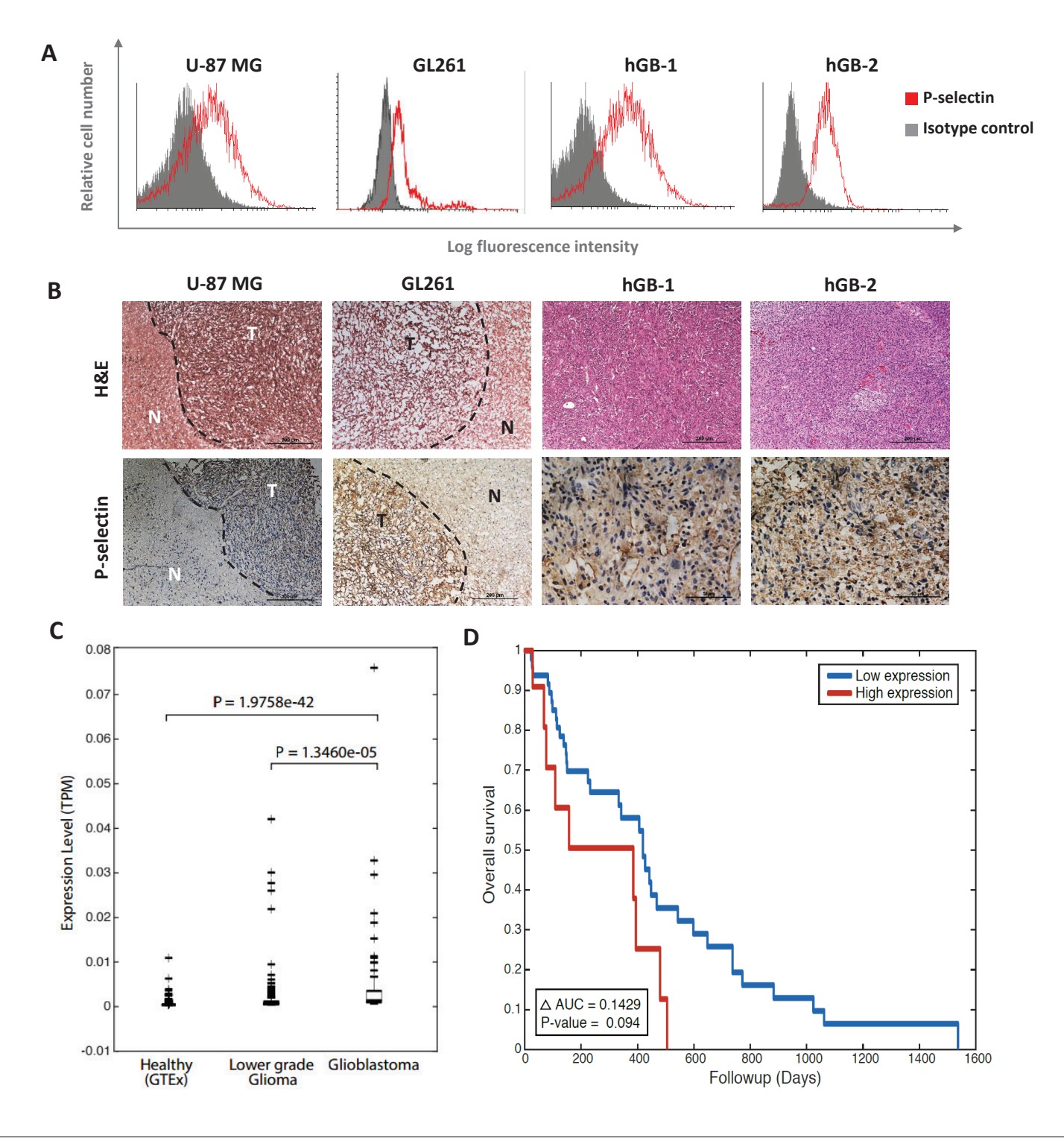

**Figure 3.** P-selectin is expressed in human and murine glioblastoma. (**A**) Flow cytometry analysis of P-selectin expression in U-87 MG, GL261 and two patient-derived glioblastoma cell lines. Images are representative of 3 individual experiments. (**B**) Representative H&E staining and immunohistochemistry staining for P-selectin expression in intracranial U-87 MG and GL261 tumors and in glioblastoma patient specimens. Positive staining is shown in brown. hGB-1 and hGB-2: human glioblastoma patient-derived cells and tissues. T: Tumor; N: Normal. (**C**) Comparison of P-selectin expression in healthy brain, lower grade gliomas and glioblastoma from data obtained from the Genotype-Tissue Expression (GTEx) collection and from The Cancer Genome Atlas (TCGA). (**D**) Kaplan-Meier survival curves obtained from TCGA data of glioblastoma patients with high and low P-selectin expression (using 63 samples with top and bottom 10% of SELP expression).

*Figure 3 continued on next page*

*Figure 3 continued*

DOI: https://doi.org/10.7554/eLife.25281.008

The following figure supplements are available for figure 3:

**Figure supplement 1.** P-selectin (SELP) is differentially expressed across various cancer types.

DOI: https://doi.org/10.7554/eLife.25281.009

**Figure supplement 2.** SELP expression in healthy and cancerous tissues.

DOI: https://doi.org/10.7554/eLife.25281.010

To validate these findings, we looked at The Cancer Genome Atlas (TCGA) data analyzing the expression of P-selectin (SELP) in glioblastoma as well as other cancer types. P-selectin expression was highly variable in all tumor types, but was the lowest in glioblastoma (*Figure 3—figure supplement 1*). However, as the expression of a gene in a given cancer should be compared to its levels in the relevant healthy reference tissue, such a comparison across different tumor types is less relevant. Indeed, when looking at SELP expression in different healthy tissues in the Genotype-Tissue Expression (GTEx) collection, we found that its levels are lowest in the brain (data not shown). We hence compared the expression levels of SELP gene in brain tissues obtained from three studies: healthy brain tissue from GTEx data (1259 samples) and lower grade glioma and glioblastoma from TCGA obtained from cBioPortal (530 and 578 samples, respectively). The expression levels of P-selectin were found to be markedly increased in gliomas compared to healthy brain tissue and further increased in glioblastomas. The resulting medians are 3.3234e-05, 1.8100e-04 and 9.4268e-04 for healthy tissue, lower grade glioma and glioblastoma, respectively (*Figure 3C*). These data suggest that even though P-selectin expression in glioblastoma is low compared to other tumor types, it represents a suitable target for glioblastoma therapy due to its potential for achieving effective tumor targeting, while minimizing side effects. This finding is of great significance in glioblastoma, a disease which represents an urgent unmet clinical need. While analyzing the ratio of P-selectin's mean expression in cancerous tissues *versus* its expression in their corresponding organs, we found that one of the most pronounced ratios received was between glioblastoma and healthy brain. Of particular interest were pancreatic cancer (PAAD) and clear cell renal cell carcinoma (ccRCC), which also seem like attractive targets for P-selectin-targeted therapies (*Figure 3—figure supplement 2*).

Having demonstrated that P-selectin is overexpressed in glioblastoma patients, we aimed to evaluate its role in glioblastoma progression. A gene-expression-based survival analysis using TCGA data obtained from cBioPortal showed that P-selectin expression correlated with survival of glioblastoma patients (*Figure 3D*).

## In vitro anti-tumor and anti-angiogenic effects of dPGS-PTX

PTX administration to cells results in the formation of microtubule bundles, arresting cells during mitosis. To evaluate PTX's ability to induce cell cycle arrest following conjugation to the dendritic nanocarriers, we performed cell cycle analysis of U-87 MG cells after treatment with PTX, dPG-PTX or dPGS-PTX. A time-dependent increase in G2/M population was observed in all treatment groups, indicating that the conjugates induce cell cycle arrest at the G2/M phase (*Figure 4A*). These results suggest that PTX retains its anti-mitotic activity both in dPGS-PTX and dPG-PTX conjugates. Confocal images of cells following 8 hr incubation in the presence of dPGS-PTX show formation of microtubule bundles right before mitosis in a large population of the cells (*Figure 4B*).

Next, we assessed the in vitro cytotoxic activity of dPG-PTX and dPGS-PTX nanoconjugates. With the aim of targeting both cancer cells and tumor vasculature, human glioblastoma cells, U-87 MG and U251, as well as endothelial cells were incubated in the presence of dPGS-PTX, dPG-PTX or free PTX at equivalent PTX dose for a prolonged period of time, allowing complete release of the drug (*Sousa-Herves et al., 2015*). Seventy-two hours following incubation, both dendritic conjugates induced cell death at a similar extent as the free drug in all cell types tested; U-87 MG and U251 glioblastoma cells and primary human umbilical vein endothelial cells (HUVEC) (*Figure 4C–E*). Thus, we concluded that PTX conjugation to dPGS provides targeting to glioblastoma cells and results in efficient internalization into P-Selectin expressing cells, while maintaining PTX anti-tumorigenic and anti-angiogenic properties.

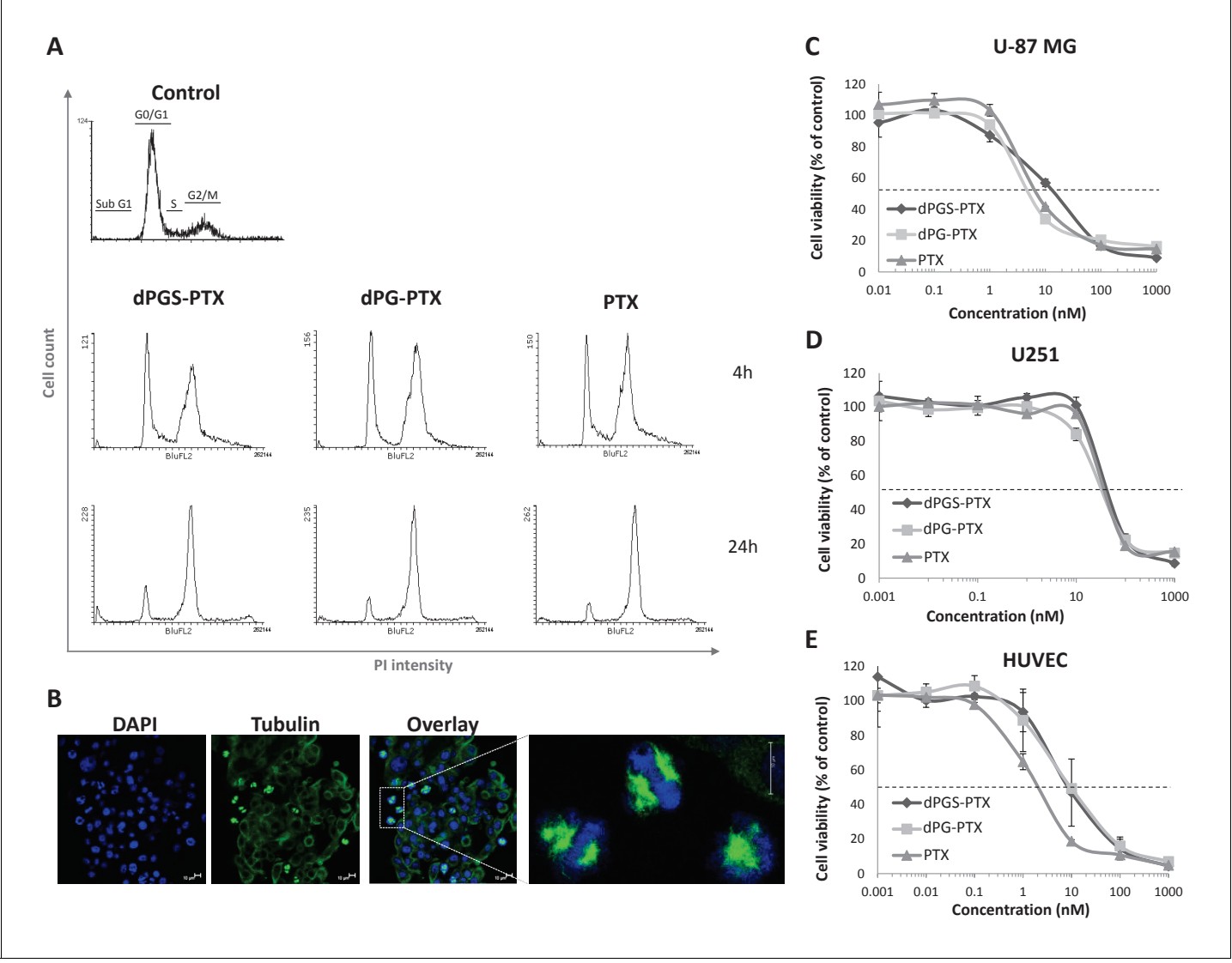

**Figure 4.** The anti-proliferative activity of PTX is retained following conjugation with dPG or dPGS nanocarriers. (**A**) Flow cytometry analysis of cell cycle following treatment with dPG-PTX, dPGS-PTX or free PTX at different time points. Images are representative of 3 individual experiments. (**B**) Representative confocal images of cells treated with dPGS-PTX for 8 hr. The nucleus is stained with DAPI (blue) and tubulin is stained with FITC-labeled antibody (green). (**C–E**) U-87 MG and U251 human glioblastoma cells and human umbilical vein endothelial cells (HUVEC) were incubated with serial concentrations of dPG-PTX, dPGS-PTX or free PTX for 72 hr and growth inhibition was evaluated. Data represent mean ± s.d. of triplicate wells. Graphs are representative of 3 individual experiments.

DOI: https://doi.org/10.7554/eLife.25281.011

The following source data and figure supplements are available for figure 4:

**Source data 1.** Raw data of U-87 MG, U251 and HUVEC proliferation assays.
DOI: https://doi.org/10.7554/eLife.25281.013

**Figure supplement 1.** Differences in the in vitro activity of the dendritic conjugates are attributed to differences in their internalization kinetics.
DOI: https://doi.org/10.7554/eLife.25281.012

**Figure supplement 1—source data 1.** Raw data of U-87 MG proliferation assay following short-term exposure to treatments.
DOI: https://doi.org/10.7554/eLife.25281.014

To note, these results were received when the cells were bathed for 72 hr with medium containing both compounds. Within 72 hr, most nanomedicines will internalize via endocytosis, whether it is fluid-phase pinocytosis (for the non-targeted conjugate) or receptor-mediated endocytosis (for the P-selectin-targeted conjugate). Therefore, we do not expect to see any differences in activity

between the targeted and non-targeted dendritic conjugates in a standard cytotoxicity assay. To that end, we ran a pulse and chase assay where we treated U-87 MG glioblastoma cells with free PTX (which diffuses the fastest into the cells), with dPGS-PTX (which internalizes rapidly by receptor-mediated endocytosis) or by dPG-PTX (which passively internalizes via fluid-phase pinocytosis) leaving the cells for 72 hr and reading their viability thereafter. It can be seen that, as expected from these three compounds, there was a difference in $IC_{50}$ exhibiting the lowest $IC_{50}$ for free PTX (100 nM), intermediate for dPGS-PTX (400 nM) and the highest for the non-targeted dPG-PTX (N/A). This explains that even in vitro, we can detect differences in activity due to differences in internalization kinetics. The phenomenon is expected to be greatly enhanced in real in vivo settings when the compounds are flowing in the bloodstream and extravasating through the tumor leaky vessels, binding to cells expressing P-selectin (*Figure 4—figure supplement 1*).

## In vivo intracranial tumor targeting of dPGS-PTX

Therapeutic effect of glioma in general, and glioblastoma in particular, is often limited by low permeability of delivery systems across the BBB and poor penetration into the tumor tissue. dPGS conjugates have been shown previously to target P/L-selectin expressed at inflamed tissues (*Dernedde et al., 2010*; *Sousa-Herves et al., 2015*; *Weinhart et al., 2011b*). Having shown preferable internalization of dPGS into glioblastoma cells compared to dPG in vitro, we proceeded with evaluating the ability of dPGS-PTX to overcome these two barriers in vivo as it is known that one of the cancer hallmarks is inflammation (*Hanahan and Weinberg, 2011*). The distribution and targeting capability of dye-labeled dPGS-PTX-IDCC was studied by intravital fluorescence imaging of mice bearing intracranial mCherry-labeled U-87 MG tumors. As shown in *Figure 5A*, one hour following systemic administration, dPGS-PTX-IDCC evidently accumulated at the mCherry-labeled tumor. Even with a higher exposure time, no IDCC signal was observed elsewhere in the body and the surrounding CNS tissue (IDCC signal was validated by spectral unmixing). Twenty-four hours later, substantial amount of the conjugate was still located at the intracranial tumor, as exhibited by a strong fluorescence signal (*Figure 5A*). Interestingly, only a faint signal was detected at the tumor core. This can be explained by the presence of a necrotic core within the tumor mass, which is a characteristic of high-grade gliomas (*Gudinaviciene et al., 2004*), that prevents the conjugate from penetrating. Another explanation is that dPGS-PTX has limited infiltration throughout the tumor tissue and therefore localize mainly at outer tumor areas. It is important to note that all glioblastoma patients are treated following tumor resection for residual tumor cells. Consequently, limited tumor penetration by dPGS-PTX may not be an impediment. Preferable accumulation at the tumor site was not observed following systemic delivery of dPG-PTX, which relies upon passive extravasation-dependent targeting alone (*Figure 5A*). Though, intracranial localization was observed to some extent. In fact, the strongest fluorescence signal was observed at what seems to be the liver location (according to fluorescence location), similar to many other DDS. Twenty-four hours following systemic administration, our control dPG-PTX is still observed in the brain, but at other organs at similar extent. These results suggest that when administered systemically, our sulfated conjugate will cross the BBB, selectively accumulate at the intracranial tumor site, and release the active drug.

Confocal imaging of brain sections corroborated the targeting capabilities of the sulfated dendritic conjugate both in U-87 MG and in GL261 tumor xenografts. One hour following injection, dPGS-PTX-IDCC was clearly visualized within the tumor tissue but not in the normal brain (*Figure 5B*). At 24 hr following administration, dPGS-PTX-IDCC was still evident, though to a lesser extent, inside the tumor (*Figure 5—figure supplement 1A*). Conversely, the localization of dPG-PTX-IDCC in the tumor was negligent at both time points, with a slight increase of IDCC signal in the normal brain compared to dPGS-PTX-IDCC (*Figure 5B*, *Figure 5—figure supplement 1B*). To understand where dPGS-PTX-IDCC acts within the tumor, we stained U-87 MG and GL261 tumor sections for P-selectin and CD31. As expected, P-selectin was highly expressed in both tumor types. However, although the IDCC dye highly colocalized with P-selectin in the GL261 tumors, its presence was less evident in all P-selectin positive cells in U-87 MG tumors (*Figure 5C*). Conversely, a clear co-localization between dPGS-PTX-IDCC and the tumor endothelium was shown (*Figure 5D*). This suggests that P-selectin-targeted dPGS-PTX works primarily on P-selectin-expressing tumor vasculature and only then penetrates the tumor and affects P-selectin-expressing glioblastoma cells.

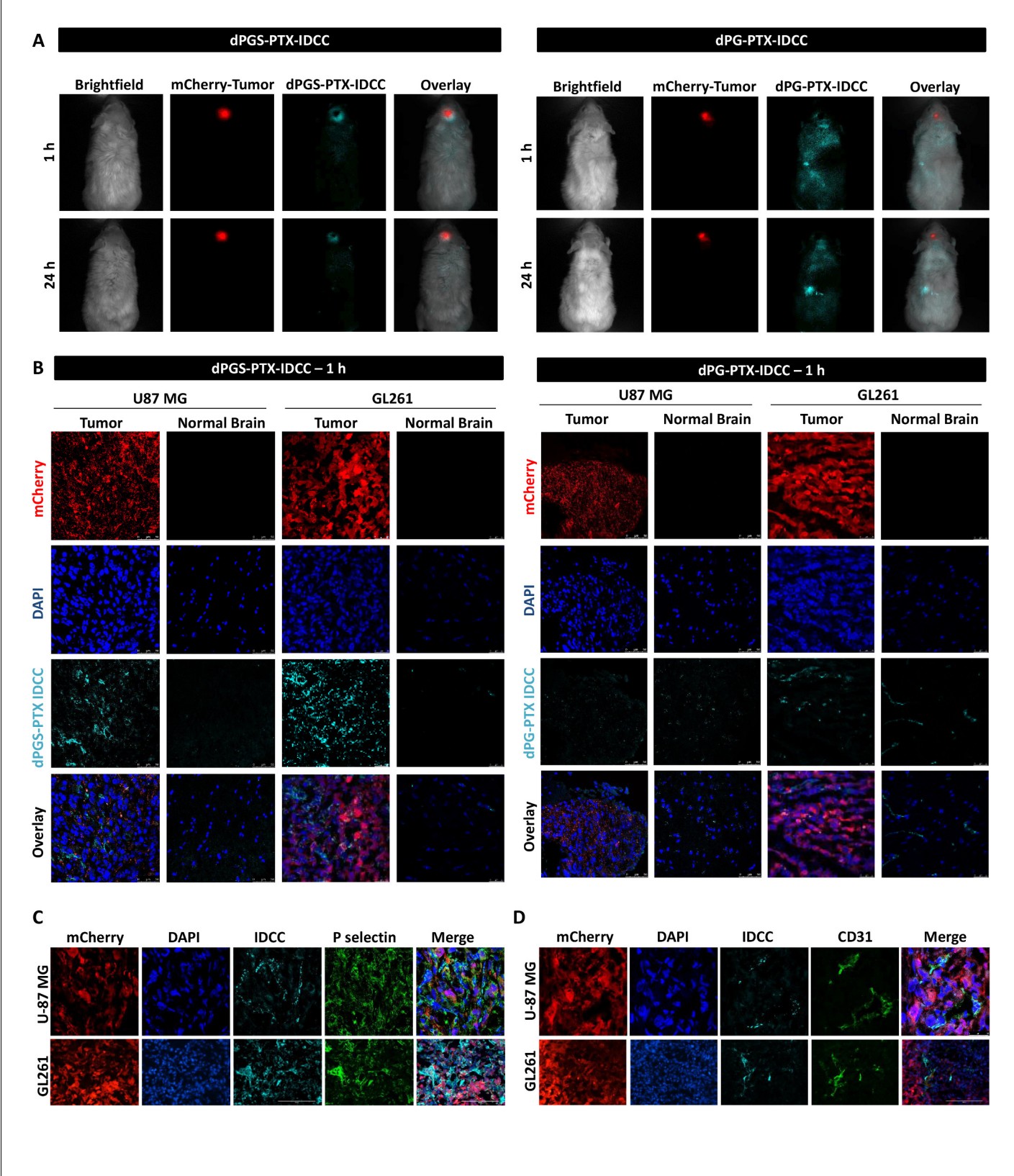

**Figure 5.** dPGS-PTX preferably accumulates in intracranial tumors. (A) Representative non-invasive fluorescence images of mice bearing mCherry-labeled U-87 MG tumors (red) 1 hr and 24 hr following intravenous injection of IDCC-labeled dendritic conjugates (cyan) (n = 2). (B). Representative

*Figure 5 continued on next page*

*Figure 5 continued*

confocal images of brain sections of mice bearing U-87 MG or GL261 tumors 1 hr following intravenous injection of the dendritic conjugates (n = 2–3). Images depict DAPI-stained nucleus (blue), mCherry-labeled tumor cells (red) and IDCC-labeled dendritic conjugates (cyan). (C–D) Representative immunohistochemical staining for P-selectin (C) or CD31 (D) in U-87 MG and GL261 tumors following administration of dPGS-PTX-IDCC. Positive staining is shown in green (n = 3).

DOI: https://doi.org/10.7554/eLife.25281.015

The following figure supplement is available for figure 5:

**Figure supplement 1.** dPGS-PTX preferably accumulates in intracranial tumors at 24 hr.

DOI: https://doi.org/10.7554/eLife.25281.016

## Thrombospondin-1 mimetic peptide (TSP-1 PM) has an anti-angiogenic effect in glioblastoma

As PTX-related toxicity is dose-dependent, we proposed to exploit combination therapy to investigate the possibility of an additive therapeutic effect while aiming to reduce the total required dose of PTX.

Therefore, we next aimed to evaluate the anti-tumor effect of dPGS-PTX in combination with a potent angiogenesis inhibitor. To do so, we utilized TSP-1 PM that induces anti-angiogenic activity by binding to TSR ligands (*Henkin and Volpert, 2011*). This TSP-1 PM is an octapeptide (*Figure 6A*), with high potency and low toxicity (*Haviv et al., 2005*; *Volpert et al., 1995*). Thus, the ability of TSP-1 PM to inhibit the formation of new blood vessels from an ex vivo aortic ring was evaluated (*Figure 6B*). An aortic ring was resected from mice, placed in matrigel and incubated in media from confluent U-87 MG cells. Extensive proliferation, sprouting and formation of tubular structures of endothelial cells were observed 8 days following incubation in media containing the pool of factors secreted by U-87 MG cells. However, low levels of TSP-1 PM (1 ng/mL) were sufficient for complete inhibition of endothelial cells sprouting. TSP-1 PM also inhibited VEGF-induced hyperpermeability in vivo in a miles assay following 2 days of treatment (*Figure 6C–D*). Furthermore, basal vessels permeability was found to be significantly decreased in treated mice. This is in accordance with previous publications that showed inhibition of VEGF-induced hyperpemeability of tumor vasculature and the formation of new blood vessels in vivo by TSP-1 PM (*Anderson et al., 2007*; *Nakamura et al., 2012*). This suggests that TSP-1 PM is a promising candidate for our combination therapy approach, as it efficiently induces TSP-1 antiangiogenic downstream signaling pathways. To evaluate the anti-angiogenic potency of TSP-1 PM in vivo, mice bearing U-87 MG glioblastoma tumors were treated with the peptide and tumors were analyzed immunohistochemically for expression of angiogenic markers. Tumors treated with TSP-1 PM were significantly less vascularized, expressed lower levels of VEGF, and existing blood vessels were stabilized, as shown by positive staining for α smooth muscle actin (αSMA) (*Figure 6E–F*).

## Combination of TSP-1 PM with dPGS-PTX enhances the therapeutic effect of PTX

To evaluate the therapeutic efficacy of TSP-1 PM and PTX combination, we performed an annexin V/propidium iodide (PI) apoptosis assay on HUVEC. The combination treatment enhanced both early (annexin V+/PI-) and late (annexin V+/PI+) apoptosis, compared to each treatment alone (*Figure 7A*). To evaluate the anti-tumor effect of TSP-1 PM and PTX combination on several cellular compartments of glioblastoma, we established a 3D tumor spheroid model composed of patient-derived glioblastoma cells, human cerebral microvascular endothelial cells (hCMEC/D3) and human astrocytes. Glioblastoma spheroids were embedded in matrigel, added with the treatments and allowed to sprout and invade into the matrigel. Combination of TSP-1 PM and PTX inhibited both endothelial and glioblastoma cells' sprouting to a greater extent compared to the other treatment groups (*Figure 7B*).

It has been demonstrated previously that TSP-1 PM can increase the concentration of chemotherapeutic agents at the tumor site through vessel normalization (*Campbell et al., 2010*). Therefore, we set to determine whether TSP-1 PM synergizes with dPGS-PTX via a similar mechanism. Mice bearing orthotopic mCherry-labeled GL261 tumors were treated with TSP-1 PM for 7 days, followed by intravenous injection of IDCC-labeled dPGS-PTX or dPG-PTX. αSMA staining confirmed that TSP-1

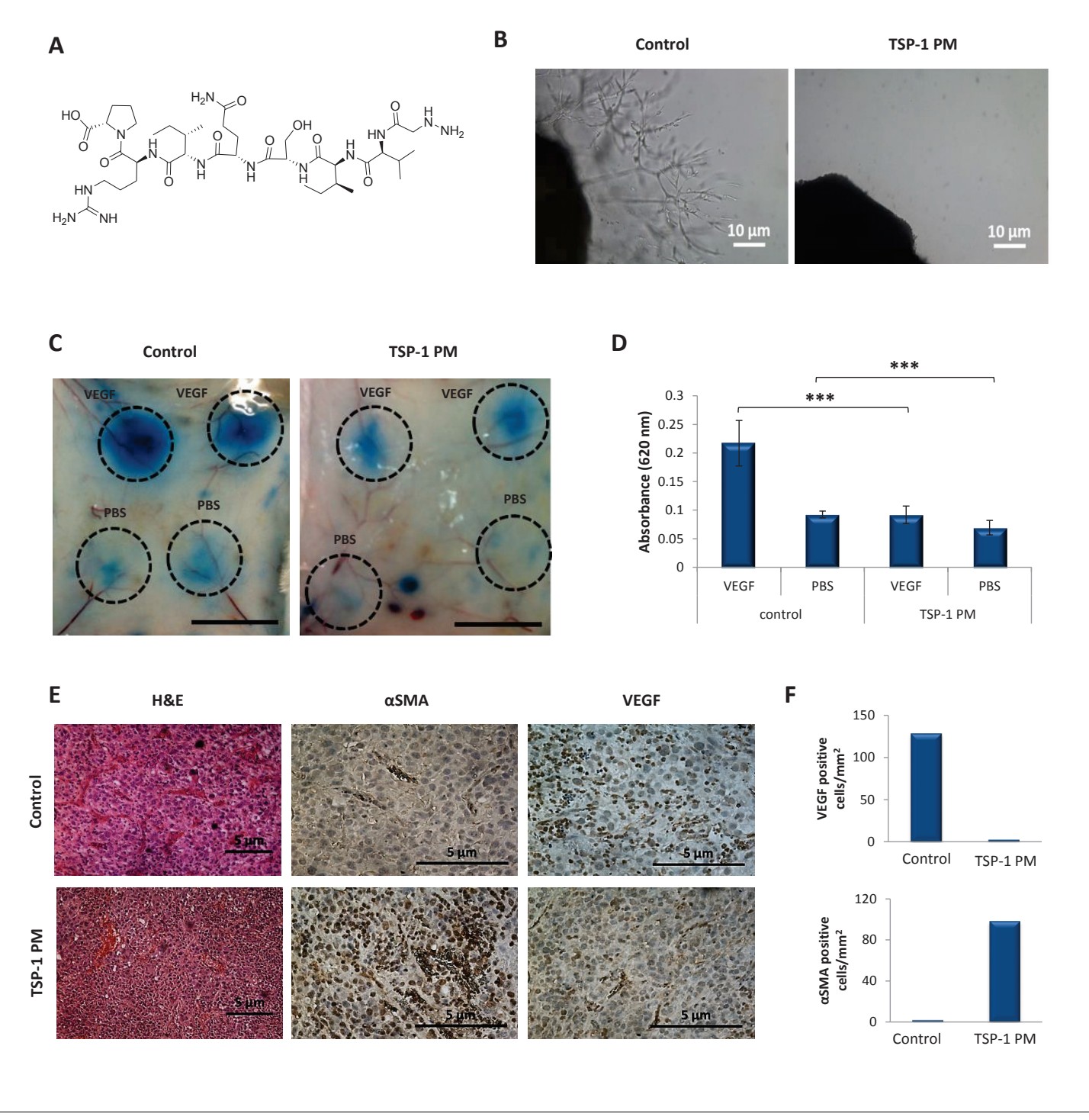

**Figure 6.** TSP-1 PM reduces the angiogenic potential of glioblastoma. (**A**) Chemical structure of ABT-898, a TSP-1 mimetic peptide. (**B**) Sprouting of endothelial cells from mouse aortic following 8 days incubation with conditioned media collected from U-87 MG cells, in the absence or presence of TSP-1 PM. Images are representative of 3 individual experiments. (**C**) Assessment of vascular permeability by Miles assay following treatment with TSP-1 PM (n = 3). (**D**) Quantification of Evans Blue dye extracted from the skin. Data represent mean ± s.e.m. ***p<0.01. (**E**) Representative immunohistochemical staining of U-87 MG tumors (n = 3). Sections were stained with H&E or stained for αSMA and VEGF. (**F**) Quantification of αSMA or VEGF positive cells within U-87 MG tumors.

DOI: https://doi.org/10.7554/eLife.25281.017

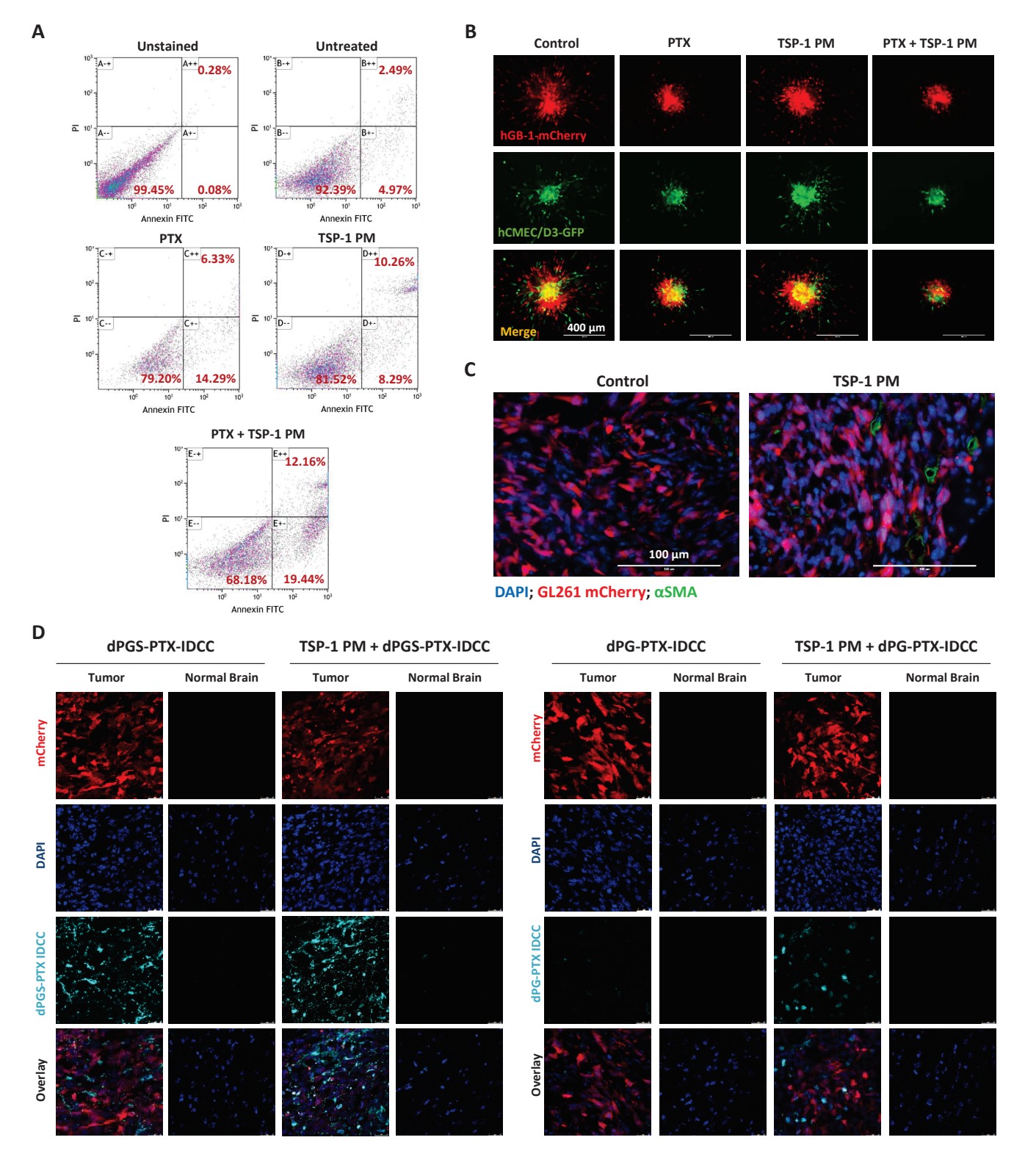

**Figure 7.** TSP-1 PM synergizes with dPGS-PTX to promote enhanced accumulation at the tumor site and cellular apoptosis. (**A**) Flow cytometry analysis of apoptosis in HUVEC following treatment with 1 ng/ml TSP-1 PM, 10 nM PTX or their combination. (**B**) Representative images of 3D spheroid invasion following treatment with 1 ng/ml TSP-1 PM, 10 nM PTX or their combination. (**C**) Representative images of αSMA staining in GL261 tumors following treatment with saline (control) or 100 mg/kg TSP-1 PM. (**D**) Representative confocal images of brain sections 24 hr following intravenous injection of the

*Figure 7 continued on next page*

Figure 7 continued

dendritic conjugates, in the presence or in the absence of a 7 day pre-treatment with TSP-1 PM (100 mg/kg; QD IP). Images depict DAPI-stained nucleus (blue), mCherry-labeled tumor cells (red) and IDCC-labeled dendritic conjugates (cyan) (n = 3).
DOI: https://doi.org/10.7554/eLife.25281.018

PM stabilizes blood vessels in GL261 tumors (*Figure 7C*), thereby enhancing the internalization of the dendritic conjugates, as demonstrated by confocal imaging of tumor sections (*Figure 7D*).

## In vivo intracranial anti-tumor efficacy of dPGS-PTX in combination with TSP-1 PM

With these promising results in hand, we set to test the anti-tumor efficacy of our combination therapy in immuno-competent mice. Previous studies have shown great similarities between orthotopic murine GL261 model and human glioblastoma (*Doblas et al., 2010*). Morphological and biochemical differences between models (*i.e*, genes involved in tumor progression, aggressiveness, infiltration to the surrounding normal brain, and angiogenic ability) could potentially result in different therapeutic outcome. Therefore, GL261 was chosen as a reliable glioblastoma model to assess the anti-tumor efficacy of the following combination therapy. C57BL/6 mice were inoculated intracranially into the striatum with GL261 cells and were allowed to establish tumors for 7 days. Mice were then randomized into groups containing 5 to 8 mice per group and administered systemically with PTX, dPG-PTX or dPGS-PTX (15 mg/kg equivalent PTX; QOD) and/or TSP-1 PM (100 mg/kg; QD IP) for 2 weeks. An additional group was treated with TMZ, the current standard of care for glioblastoma patients. Animal survival after treatment with dPGS-PTX in combination with TSP-1 PM was significantly longer compared to the untreated control group or TMZ-treated group (*Figure 8A*). Combining dPGS-PTX with TSP-1 PM was also found to provide survival benefits compared to all the groups treated with the non-targeted dPG-PTX conjugate, with or without TSP-1 PM. The benefit of our targeted conjugate and the combination therapy is nicely exhibited by a 'dose-response' in mice survival following treatment. Mice treated with combination of dPGS-PTX and TSP-1 PM exhibited a remarkable 100% survival rate, with all mice surviving more than 120 days. This was followed by 40% of mice surviving more than 120 days in the group treated with dPGS-PTX alone, 25% of mice treated with dPG-PTX and TSP-1 PM, and 10% of mice treated with dPG-PTX alone. All DDS-administered groups exhibited survival benefit over free PTX and TMZ. Interestingly, TSP-1 PM as monotherapy did not result in any survival benefit. Due to the positive interaction between dPGS-PTX and TSP-1 PM, the improvement in survival observed is highly indicative of a synergistic effect in vivo, and has a potential to serve as an alternative to the existing oral TMZ. Several studies have shown correlation between dPGS binding to selectins and inhibition of inflammation (*Dernedde et al., 2010*). In order to evaluate whether dPGS has an inhibitory effect on the immune system in a cancer model as well, and if that translates to survival benefit or impairment, mice were treated with non-PTX bearing conjugates. Both dPGS and dPG did not significantly affect mice survival compared to non-treated controls, suggesting that dPGS does not have an inhibitory effect on inflammation in a glioblastoma model to an extent that influences the therapeutic outcome.

To evaluate treatment-induced systemic toxicities, mice were followed for body weight changes and their well-being. Additionally, blood was extracted from mice for total blood count and biochemistry analysis. Body weight measurements showed that no significant weight loss was observed following any of the treatments (*Figure 8B*). Blood test results indicated that mice treated with dPGS-PTX plus TSP-1 PM did not exhibit any systemic toxicity and generally seemed healthier compared with non-treated control mice or any other treatment. In contrast, myelosuppression that is associated with TMZ treatment was observed by decreased white blood cells count in TMZ treated mice. High total bilirubin together with elevated liver enzymes (SGOT and SGPT), an indication for liver dysfunction, was shown in mice treated with free PTX and TMZ. Surprisingly, mice administered with dPG and dPGS without PTX as well as dPG-PTX in combination with TSP-1 PM, were found to have elevated creatinine levels, which might indicate kidney damage (*Figure 8—figure supplement 1A–F*). Since systemic administration of PTX is normally associated with neurotoxicity, mice were evaluated for their motor coordination 20 days following treatment initiation. Treated mice were evaluated for the time period they succeeded to stay on the rod without falling (*Figure 8—figure*

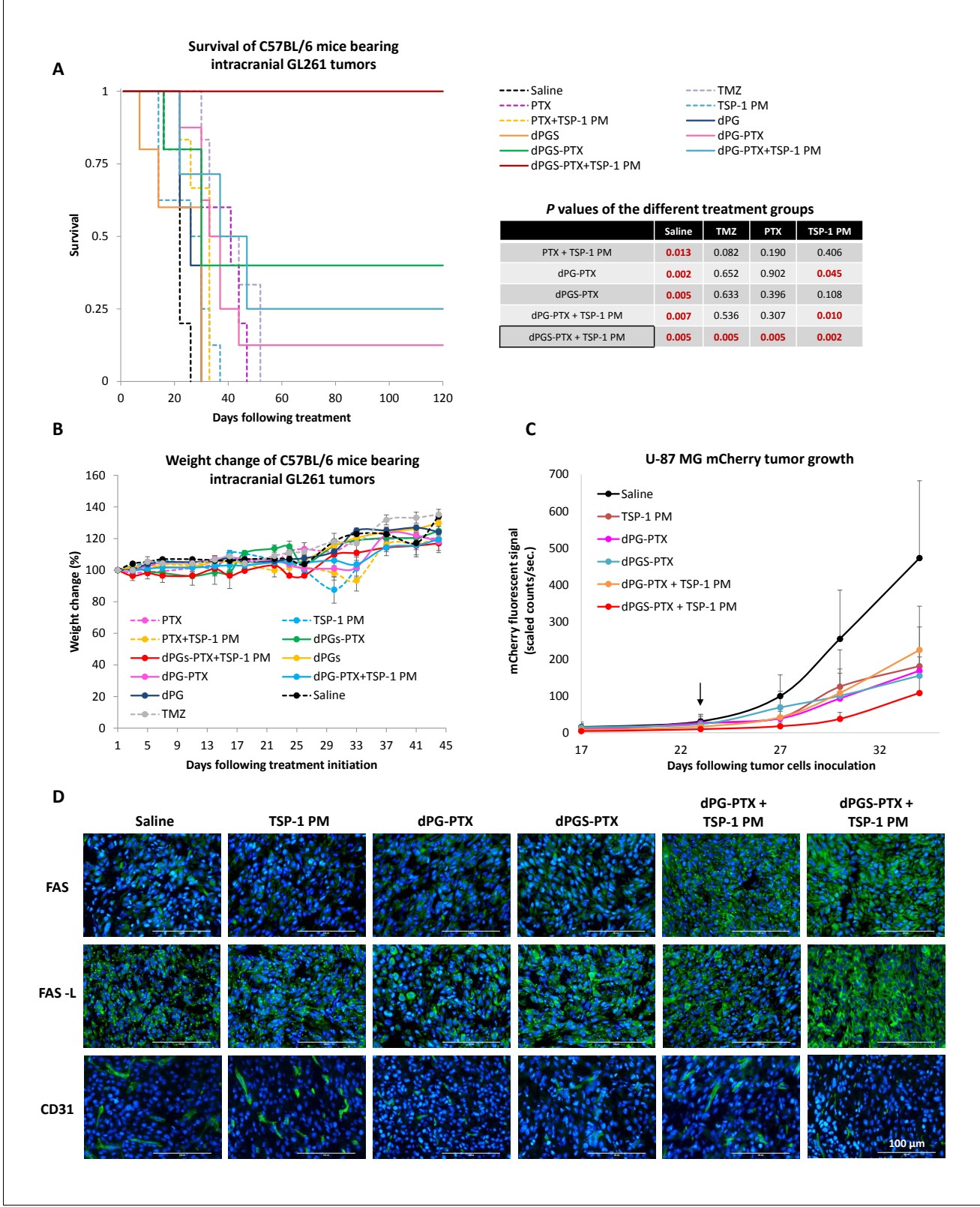

**Figure 8.** Combination therapy of dPGS-PTX with TSP-1 PM inhibits tumor growth and prolongs the survival of mice bearing intracranial glioblastoma. (**A**) Kaplan-Meier curve for survival of mice bearing murine GL261 intracranial tumors following systemic injection of treatments (n = 4–8). Table shows statistical significance between different treatment groups. P values were determined using log rank test. (**B**) Body weight change, expressed as percent change from the day of treatment initiation. Data represent mean ± s.e.m. (n = 4–8). (**C**) Tumor growth of mCherry-labeled U-87 MG tumors following

*Figure 8 continued on next page*

*Figure 8 continued*

systemic administration of treatments (n = 3–8). Arrow points to time of treatment initiation. Data is represented as scaled counts/sec. of the mCherry fluorescent signal, detected by CRI Maestro imaging system. (**D**) Representative immunohistochemical images of U-87 MG tumors treated with TSP-1 PM, dPG-PTX, dPGS-PTX or their combinations. Tissues were stained for Fas/CD95, Fas-L/CD178 and CD31. Tumor cells are shown in blue (DAPI) and positive immunostaining is shown in green (n = 3). Data represent mean ± s.e.m.

DOI: https://doi.org/10.7554/eLife.25281.019

The following source data and figure supplements are available for figure 8:

**Source data 1.** Raw data of Kaplan-Meier survival curve.
DOI: https://doi.org/10.7554/eLife.25281.021
**Source data 2.** Raw data of the fluorescent signal of mCherry labeled U-87 MG intracranial tumors.
DOI: https://doi.org/10.7554/eLife.25281.022
**Figure supplement 1.** Combination treatment of dPGS-PTX and TSP-1 PM did not induce systemic toxicity.
DOI: https://doi.org/10.7554/eLife.25281.020
**Figure supplement 1—source data 1.** Raw data of blood count and biochemistry analysis.
DOI: https://doi.org/10.7554/eLife.25281.023
**Figure supplement 1—source data 2.** Raw data of mice Rotarod experiments.
DOI: https://doi.org/10.7554/eLife.25281.024

*supplement 1G*) and their ability to learn new skills by measuring the delta between their first and last performance (*Figure 8—figure supplement 1H*). In both skills set, mice administered with the targeted sulfated dPGS-PTX conjugate exhibited the best performance with no motor deficit. The non-targeted dPG-PTX conjugate, though exhibited improved therapeutic outcome, caused neurotoxicity in treated mice similarly to the free drug. This suggests that by targeting PTX to the tumor microenvironment and the cancer cells, we were able to abrogate PTX-related neurotoxicity.

Having shown improved survival of C57BL/6 mice bearing GL261 tumors following treatment with our combination therapy, we set to evaluate its therapeutic efficacy and mechanism of synergy in an additional in vivo tumor model. As we wanted to mimic the clinical setting, we designed a short intervention study initiated at a large tumor size that allows for the observation of our treatment's effect on clinically-relevant rapidly-growing glioblastoma. SCID mice were inoculated intracranially into the striatum with mCherry labeled U-87 MG cells and tumor growth was monitored using an intravital non-invasive fluorescence imaging system (CRI Maestro). Once a sufficient fluorescence signal was visualized, mice were randomized into groups and administered systemically with dPG-PTX or dPGS-PTX (15 mg/kg equivalent PTX; QOD) and/or TSP-1 PM (100 mg/kg; QD). The experiment ended when mice from the control group were moribund and/or had a high fluorescent signal (>500 scaled counts/sec). As shown in *Figure 8C*, all treatment groups were able to inhibit U-87 MG tumor growth. However, only combination treatment of dPGS-PTX with TSP-1 PM resulted in a marked tumor growth inhibition compared to the control group. In contrast to the previous experiment with GL261, in which tumors were treated at a very early stage following tumor cells inoculation, U-87 MG tumors were treated at a relatively late stage. Therefore, it is not surprising that differences were seen in the treatment outcome between these experiments. To understand by which mechanism dPGS-PTX and TSP-1 PM synergize in vivo, the effect of the treatments on the Fas/Fas-ligand (Fas-L) apoptosis pathway was analyzed immunohistochemically. Results demonstrate that expression of Fas in the tumor tissue was enhanced following combination of TSP-1 PM with either dPG-PTX or dPGS-PTX. Fas-L expression was evident in all treatment groups. This is supported by a previous report, which showed high Fas-L expression in aggressive intracranial malignancies (*Yu et al., 2003*). Nonetheless, a marked increase in Fas-L expression was demonstrated following treatment with combination of TSP-1 PM and dPGS-PTX. Furthermore, decreased CD31 staining was demonstrated in tumors treated with our dendritic conjugates (*Figure 8D*).

## Discussion

Malignant gliomas are the most common primary brain tumors and typically widely disseminate within the normal brain. Glioma cells can be found several centimeters away from the main tumor mass and give rise to tumor recurrence after resection or radiation. The main goal of glioblastoma treatment is maximal resection with preservation of neurological function. This is facilitated by

several pre- and intra-operative evaluations of functional neuro-anatomy using advanced imaging modalities and imaging with fluorescent tumor markers (*e.g.,* ALA, 5-aminolevulinic acid) (*Widhalm, 2014*). Patients are then treated for residual disease with a combination of radiotherapy and the alkylating agent TMZ. One major factor that contributes to poor treatment outcomes of glioblastoma is a high degree of resistance to DNA alkylation-based chemotherapy. The resistant phenotype is triggered by the activity of the O6-methylguanine-DNA methyltransferase (MGMT) DNA-repair enzyme that blunts the therapeutic effect of alkylating agents. The identification of MGMT as the key player in the resistance to chemotherapy in glioblastoma, and the fact that it serves not only as prognostic but also as predictive factor, have made this enzyme a potential target for personalized therapy and facilitated the development of several inhibitors (*Cancer Genome Atlas Research Network, 2008*). Unfortunately, clinical trials have shown disappointing results with severe systemic toxicity (*Quinn et al., 2009a*; *Quinn et al., 2009b*). This is presumably due to low MGMT activity in the normal brain and hematopoietic tissues. To date, there is no useful therapeutic alternative for alkylating agents in glioblastoma patients with high MGMT activity.

The mitotic inhibitor PTX represents a potential alternative chemotherapeutic agent for glioblastoma, as its potency does not depend on the cells' MGMT status. However, its clinical use is impeded due to its poor brain tumor delivery and to its high toxicity, mainly neurological. Several delivery systems bearing PTX have been developed and studied in order to reduce PTX-related toxicity. One approach exploits nanocarriers to safely deliver PTX via the circulation. In the study presented here, we synthesized dPGS loaded with PTX for glioblastoma therapy. dPGS was conjugated to PTX via a pH-sensitive EMCH linker. We have previously conjugated dPGS to PTX through a labile ester linker that can be cleaved by esterases and/or by low pH found in the lysosome. Although this conjugate was successfully taken-up by cancer cells and had a cytotoxic effect on cancer cells growth, it demonstrated poor stability in plasma and at physiological pH (*Sousa-Herves et al., 2015*). The hydrazone linker used here improves the therapeutic properties of dPGS-PTX conjugate, as it is more stable in physiological pH and easily hydrolyzed at lower pH levels found in the tumor microenvironment or in lysosomes (*Chang et al., 2016*). This linker enabled the successful systemic delivery of dPGS-PTX to glioblastoma. Systemic administration of our dendritic conjugate significantly inhibited tumor growth of intracranial glioblastoma, circumventing the side effects related with PTX. Similar approaches for PTX delivery were recently reported using various delivery systems. PTX-loaded PEGylated poly(ε-caprolactone) nanoparticles were demonstrated to inhibit tumor growth in mice bearing intracranial glioblastoma and prolong their survival (*Xin et al., 2010*). Another approach concerns local delivery of PTX loaded nanoparticles, similarly to carmustine implants. A hydrogel matrix entrapping PLGA-PTX microspheres were synthesized by Ranganath and colleagues. This formulation demonstrated a sustained drug release and an enhanced anti-tumor efficacy when implanted into subcutaneous glioblastoma tumors (*Ranganath et al., 2009*). Chlorotoxin-targeted iron oxide nanoparticles, carrying PTX, have been shown to deliver PTX to glioma cells in vitro but no safe, effective in vivo targeting has yet been demonstrated (*Mu et al., 2015*). Although these studies did not demonstrate reduced cytotoxicity of PTX when loaded into nanocarriers, they suggest that novel strategies for PTX delivery can be promising candidates for glioblastoma therapy.

One of the major obstacles in delivering drugs to brain tumors is restricted ability to cross the BBB. We demonstrate here that penetration of both the BBB and the blood-tumor barrier is achieved by extravasation-dependent 'passive' and ligand-targeted 'active' targeting through binding of sulfate groups on the dendritic conjugate to P-/L-selectin. This barrier crossing is attributed to the expression of P-selectin on activated tumor endothelial cells (*Barthel et al., 2007*; *Läubli and Borsig, 2010*; *Shamay et al., 2016a*), as binding of the dendritic conjugate to the endothelium facilitates its extravasation via blood vessels to the target site (*Ofek et al., 2016*). Realizing the importance of selectins in cancer development and their potential for therapeutic targeting, several selectin-directed delivery systems were recently synthesized. Shamay and colleagues demonstrated an anti-tumor activity of N-(2-hydroxypropyl)methacrylamide (HPMA) copolymer conjugated to an E-selectin binding peptide and loaded with different anti-cancer agents (*Shamay et al., 2016b*; *Shamay et al., 2015*). Another P-selectin targeted drug delivery system was designed by encapsulating chemotherapies with fucoidan (Fi), an algae-derived polysaccharide with an affinity to P-selectin. They found increased expression of P-selectin in multiple cancer types including lung, ovarian, lymphoma and breast, both on tumor cells and in tumor endothelium. They further demonstrated

anti-tumor efficacious activity of their nanoparticles in melanoma, breast cancer and colon cancer mouse models (*Shamay et al., 2016a*). Here we report that P-selectin is also expressed on human glioblastoma cells and show that its expression is increased in gliomas compared to healthy brain tissues, enabling additional active targeting of human glioblastoma. To the best of our knowledge, this is the first report demonstrating increased expression of P-selectins in glioblastoma cells, rather than on glioblastoma tumors' endothelium (being increased after radiation). This emphasized the potential value of P-selectin-directed delivery of chemotherapeutics in glioblastoma, since it allows targeting the tumor and its microenvironment. It is also reported that radiation rapidly increases endothelial cell display of P-selectin on vascular lumen surfaces in a microtubule-dependent process (*Hallahan and Virudachalam, 1999*), thus sensitivity to dPGS-PTX may be further enhanced following local radiation treatment of glioma. Surprisingly, a TCGA data analysis shown here has further demonstrated that high expression of P-selectin correlates with poor patient survival. This suggests that targeting P-selectin in glioblastoma may potentially have an additional therapeutic benefit. Nevertheless, the precise cellular target of our P-selectin-targeted dPGS-PTX remains unclear and warrants further investigation. Future studies using P-selectin knockout tumor and endothelial cells may resolve this issue and uncover which of these cellular compartments is targeted by dPGS-PTX.

Another way for reducing PTX-induced toxicity is by exploiting the combinational therapy approach. PTX-associated neurotoxicity is avoided here by achieving a synergistic therapeutic effect in vivo by co-delivery with TSP-1 peptidomimetic, decreasing the necessary dose of PTX, thus allowing its accumulation at levels considerably lower than the toxic dose. TSP-1 is a potent endogenous angiogenesis inhibitor, whose expression is often lost during glioma malignant transformation. We have previously shown a correlation between loss of TSP-1 expression and escape from dormancy in human glioblastoma (*Satchi-Fainaro et al., 2012*). Thus, glioma stimulation of local endothelium should be mitigated by restoration of TSP-1 function. TSP-1 PM used here is a second generation peptide of ABT-510, which was shown to enhance the efficacy of both PTX and of cisplatin in an orthotopic syngeneic model of ovarian cancer through normalization of tumor vasculature, as documented by increased pericytes coating tumor vessels, leading to increased concentration of these agents in the tumor (*Campbell et al., 2010*). Partial decrease of lesions was changed to full remission of lesions by combining ABT-510, shown to enhance delivery. A similar mechanism of synergy was also reported when administering the anti-angiogenic agent bevacizumab prior to PTX (*Tolaney et al., 2015*). Therefore, we hypothesized that the intra-tumoral delivery of dPGS-PTX may be impeded owing to collapsed vasculature, with microcirculatory delivery similarly improved by TSP-1 PM, possibly explaining its synergistic effect. This is supported by our finding of vessel stabilization in glioblastoma and enhanced accumulation of IDCC-labeled dendritic conjugates at the tumor site following treatment with TSP-1 PM. Thus, enhanced tumor growth inhibition and long-term survival may be mainly due to vascular normalization. Alternatively, increased in vitro endothelial cell presentation of Fas (CD95) was observed with >1 nM concentrations of docetaxel, and ABT-510-dependent apoptosis of tumor endothelial cells was shown to be induced by their upregulation of Fas ligand (CD178) (*Yap et al., 2005*). We postulated that dPGS-PTX may similarly synergize TSP-1 PM by inducing the Fas/Fas-L apoptosis pathway. IHC analysis supported this hypothesis, suggesting an additional mechanism which contributes to the enhanced anti-tumor effect observed in our in vivo studies. Both of the putative mechanisms may operate and be mutually amplifying.

In conclusion, the work described here demonstrates the tremendous potential of our novel combination of TSP-1 PM and dPGS-PTX in replacing the conventional therapy for those patients who will not benefit from alkylating agents (i.e., high MGMT activity), with decreased toxicity and increased safety profile.

## Materials and methods

### Materials

DMEM, fetal bovine serum (FBS), penicillin and streptomycin were purchased from Biological Industries Ltd. (Kibbutz Beit Haemek, Israel). EGM-2 medium was from Cambrex, USA and endothelial cells growth supplement (ECGS) was from Biomedical Technologies Inc. (Stoughton, MA, USA). All other chemical reagents, including salts and solvents, were purchased from Sigma-Aldrich (Rehovot, Israel). ABT-898, TSP-1 PM, was synthesized by standard solid state methods as previously described

(*Haviv and Bradley, 2006*). Chemicals and solvents were either AR grade or purified by standard techniques. Chemicals and reagents were obtained from Acros Organics, Sigma-Aldrich, and Merck. They were reagent grade and used as received unless otherwise stated. Milli-Q water was prepared using a Millipore water purification system. Purification by centrifugal filtration was performed using Amicon Ultra Centrifugal Filters [molecular weight cut-off (MWCO) 5 or 3 KDa Millipore]. Ultrafiltration was performed on stirred cells with Amicon membranes (MWCO 5 KDa, Millipore). Size exclusion chromatography (SEC) was performed with Sephadex G-25 superfine (GE Healthcare) under ambient pressure and temperature. 2S-IDCC-maleimide dye was used as previously reported (*Krüger et al., 2014*). $^1$H NMR spectra were recorded on a Jeol ECX 400, Bruker AMX 500, or on a Bruker BioSpin AV 700 spectrometer. Chemical shifts are reported in ppm (δ units). For ESI measurements, a TSQ 7000 (Finnigan Mat) instrument was used. Elemental analysis was performed on a Vario EL III elemental analyzer using sulfanilic acid as standard. Absorption spectra were recorded on a LAMBDA 950 UV/Vis/NIR spectrometer (PerkinElmer, USA).

## Synthesis of the dendritic conjugates

### Synthesis of dPGS amine
dPGS containing free amine groups (Mn ≈ 11.6 kDa, degree of sulfation 90%, degree of amination 5%) was prepared as recently reported (*Gröger et al., 2013*). Briefly, dPG containing N-phthalimide protected amine functionalities was synthesized via the copolymerization of glycidol and *N*-(2,3-epoxypropyl)phthalimide on a partly deprotonated trimethylol propane (TMP) starter. Sulfation of the hydroxyl groups, followed by cleavage of the N-phthalimide protecting groups, yielded dPGS with free amine functionalities available for further conjugation. The sulfur content was determined by elemental analysis and corresponded with a degree of sulfation of 90%.

### Synthesis of dPG amine
dPG amine ($M_n$ ≈ 10 KDa, 10% amine groups) was synthesized according to previously reported procedures (*Ofek et al., 2016*). In brief, dPG ($M_n$ ≈ 10 kDa) was prepared following literature procedures (*Sunder et al., 2000*) and then 10% of the hydroxyl groups were converted into amine functionalities by means of a three step protocol. First, dPG was treated with methanesulfonyl chloride, and the resulting mesyl groups were substituted by azides by reaction with NaN$_3$ in DMF. Finally, reduction of the azide groups with PPh$_3$ rendered dPG amine. Extensive dialysis was carried out after each reaction step and quantification of the % of NH$_2$ groups was performed using $^1$H NMR spectroscopy.

### Synthesis of the PTX-EMCH
An ester-derivative of PTX at the C-2'-OH-position (PTX-Bz) was first prepared by reaction with 4-acetylbenzoic acid following a literature report (*Rodrigues et al., 2003*). Subsequently, PTX-Bz (50 mg, 0.049 mmol) was dissolved in absolute EtOH (0.6 mL) and EMCH trifluoroacetic acid salt (33.92 mg, 0.099 mmol) was added under Ar atmosphere. The mixture was stirred at room temperature (rt) for 2 hr protected from light. The solution was then allowed to precipitate at 4°C overnight. The precipitate was centrifuged and washed with ice-cold Et$_2$O. After solvent removal, PTX-EMCH was obtained as a white solid (55 mg, 0.046 mmol, 93%). The compound was characterized by $^1$H-NMR spectroscopy and ESI-ToF mass spectrometry.

$^1$H NMR (400 MHz, DMSO-d$_6$, δ): 8.02 7.24 (m, 19 hr), 6.99 (d, *J* = 13.4 Hz, 2 hr), 6.30 (s, 1 hr), 5.80 5.87 (m, 2 hr), 5.56 (d, *J* = 8.4 Hz), 5.42 (d, *J* = 6 Hz), 4.94 (m, 1 hr), 4.91 (s, 1 hr) 4.69 (s, 1 hr), 4.13 (m, 1 hr), 4.00 (m, 1 hr), 3.61 (s, 1 hr), 2.66 (s, 3 hr), 2.33 (m, 1 hr), 2.30 (s, 3 hr), 2.10 (s, 3 hr), 1.94–1.75 (m, 2 hr), 1.83 (s, 3 hr), 1.64 (m, 1 hr), 1.59 (m, 4 hr), 1.50 (s, 3 hr), 1.24 (m, 3 hr), 1.09 (m, 3 hr), 1.00 (m, 3 hr).

MS: (+ESI, MeOH) m/z = 1207.4806 [M + H]$^+$ calculated 1207.4758, 1229.4630 [M + Na]$^+$ calculated 1229.4583.

### Synthesis of dPGS-PTX
dPGS amine (100 mg, 8.6 µmol, 4 NH$_2$ groups) was dissolved in 0.5 mL phosphate buffer (PB) 10 mM (pH 7.4). 2-iminothiolane (7.1 mg, 0.52 mmol, 1.5 eq per NH$_2$ group) was added in 250 µL of PB (pH 7.4) and the reaction mixture was stirred for 1 hr at rt. Then, PTX-EMCH (15.6 mg, 12.9 mmol,

1.5 eq per dPGS) was added in 3 mL DMF and the mixture was stirred overnight at rt. After that time, the crude product was extensively ultrafiltrated (MWCO 5 KDa) with MeOH (2x), MeOH:$H_2O$ (1:1; 2x), and $H_2O$ (2x). The obtained conjugate (57 mg, 51%) was analyzed by [1]H-NMR. The estimated molar ratio between PTX and dPGS was *ca.* 1, according to the [1]H-NMR spectrum (*Figure 1—figure supplement 2*).

[1]H NMR, (700 MHz, DMSO-$d_6$, δ): 8.18–7.04 (m, 19 hr), 6.23 (s, 1 hr), 5.60–5.25 (m, 4 hr), 4.131–3.20 (m, 330 hr), 1.45 (s, 3 hr), 0.99–0.91 (m, 6 hr).

## Synthesis of dPG-PTX

dPG amine (10% $NH_2$) (100 mg, 10 µmol, 13.5 $NH_2$ groups) was dissolved in 1.2 mL of MeOH. 2-iminothiolane (30 mg, 0.21 mmol, 1.5 eq per $NH_2$ group) was dissolved in 0.6 mL of MeOH and added to the dPG amine solution. The reaction mixture was stirred for 1 hr at rt. Afterwards, a solution of PTX-EMCH (18.0 mg, 0.015 mmol, 1.5 eq per dPG amine) in 3 mL of MeOH was added and the mixture was stirred overnight at rt. After that time, the crude product was purified by extensive filtration with Amicon filters (MWCO 5 kDa), first with PB 10 mM (pH 7.4)/MeOH (10%, 1x) and then with PB 10 mM (pH 7.4) until no free PTX was observed in the filtrate by HPLC. The pure conjugate was kept frozen in PB 10 mM (pH 7.4) solution to avoid any drug release and solubility problems after freeze-drying. In order to calculate the yield, a small fraction was taken and salts were removed. [1]H NMR spectrum DMSO-$d_6$ showed that the conjugate has about 1 mol of PTX per mol of dPG- PTX (270 mg, 48%) (Fig. S2).

[1]H NMR (700 MHz, DMSO-$d_6$, δ): 8.25–7.11 (m, 19 hr), 6.30 (s, 1 hr), 5.80–5.87 (m, 2 hr), 5.56 (m, 1 hr), 5.42 (m, 1 hr), 3.90–2.90 (675H), 2.61–2.53 (m, 6 hr), 2.39–2.23 (m, 5 hr), 2.10 (s, 3 hr), 2.06–1.95 (m, 10 hr), 1.84 (s, 3 hr), 1.42 (s, 3 hr), 1.06–0.95 (m, 6 hr).

## Synthesis of dPGS-IDCC

dPGS amine (20 mg, 1.72 µmol, 4 $NH_2$-groups) was dissolved in 0.1 mL PB 50 mM (pH 7.4). 2-iminothiolane (1.76 mg, 0.012 mmol, 1.5 eq per $NH_2$ group) was added in 50 µL of PB 50 mM (pH 7.4). After 20 min reaction, 2S-IDCC-maleimide (2.36 mg, 0.002 mmol, 1.5 eq per dPGS) was added in MeOH (0.2 mL) and the reaction mixture was stirred for 2 hr. PB 10 mM (pH 7.4) (15 mL) was added and the solution was washed using Amicon filters (MWCO 3 KDa). Finally, the solution was purified by SEC using a Sephadex-G25 superfine column. The conjugate (13.0 mg) was analyzed by UV-vis spectroscopy in $H_2O$. The obtained dye-loading was 25.5 µg/mg conjugate (extinction coefficient of IDCC at 646 nm = 250000 $M^{-1}$ $cm^{-1}$) (*Licha et al., 2001*).

## Synthesis of dPG-IDCC

dPG amine (10%) (25 mg, 2.5 µmol, 13.5 $NH_2$ groups) was dissolved in 0.3 mL MeOH. 2-iminothiolane (6.9 mg, 0.05 mmol, 1.5 eq per $NH_2$ group) was dissolved in 0.15 mL of MeOH and added to the dPG amine solution. The reaction mixture was stirred for 20 min at rt. Afterwards, 2S-IDCC-maleimide (3.2 mg, 3.48 µmol, 1.4 eq per dPG amine) was added in MeOH (0.1 mL) and the reaction mixture was stirred for 3 hr. 15 mL PB 10 mM (pH 7.4) were added and the solution was washed using Amicon filters (MWCO 3 KDa). Finally, the solution was purified by SEC. The conjugate (34 mg) was analyzed by UV-vis spectroscopy in $H_2O$. The obtained dye-loading was 5.5 µg/mg conjugate (extinction coefficient of IDCC at 646 nm = 250000 $M^{-1}$ $cm^{-1}$).

## Synthesis of dPGS-PTX-IDCC

dPGS amine (20 mg, 1.72 µmol, 4 $NH_2$ groups) was dissolved in 0.1 mL PB 50 mM (pH 7.4). 2-iminothiolane (1.76 mg, 0.012 mmol, 1.5 eq per $NH_2$ group) was added in 50 µL of PB 50 mM (pH 7.4). After 20 min reaction PTX-EMCH (3.12 mg, 0.002 mmol, 1.5 eq per dPGS) was added in 0.6 mL DMF and the mixture was stirred 30 min at rt. 2S-IDCC-maleimide (2.36 mg, 0.002 mmol, 1.5 eq per dPGS) was added in MeOH (0.2 mL) and the reaction mixture was stirred for 2 hr. 15 mL PB 10 mM (pH 7.4) (15 mL) were added and the solution was washed using Amicon filters (MWCO 3 KDa) until no free PTX was observed in the filtrate by HPLC. Finally, the solution was purified by SEC. The conjugate (18.7 mg) was analyzed by UV-vis spectroscopy in $H_2O$. The obtained dye-loading was 25.5 µg/mg conjugate (extinction coefficient of IDCC at 646 nm = 250000 $M^{-1}$ $cm^{-1}$). A 1:1 molar ratio IDCC/PTX was assumed on the basis of previous works (*Baabur-Cohen et al., 2017*).

## Synthesis of dPG-PTX-IDCC

dPG amine (10% NH$_2$) (25 mg, 2.5 µmol, 13.5 NH$_2$ groups) was dissolved in 0.3 mL MeOH. 2-imino-thiolane (6.9 mg, 0.05 mmol, 1.5 eq per NH$_2$ group) was dissolved in 0.15 mL of MeOH and added to the dPG-NH$_2$ solution. The reaction mixture was stirred for 20 min at rt. Afterwards, a solution of PTX-EMCH (4.5 mg, 3.75 µmol, 1.5 eq per dPG amine) in 0.25 mL of MeOH was added and the mixture was stirred for 20 min. Finally, 2S-IDCC-maleimide (3.4 mg, 3.75 µmol, 1.5 eq per dPG-NH$_{2)}$ was added in MeOH (0.1 mL) and the reaction mixture was stirred for 3 hr. 15 mL PB 10 mM (pH 7.4) were added and the solution was washed using Amicon filters (MWCO 3 KDa) until no free PTX was observed in the filtrate by HPLC. Finally, the solution was purified by SEC using a Sephadex-G25 superfine column. The conjugate (38 mg) was analyzed by UV-vis spectroscopy in H$_2$O. The obtained dye-loading was 24 µg/mg conjugate (extinction coefficient of IDCC at 646 nm = 250000 M$^{-1}$cm$^{-1}$). A 1:1 molar ratio IDCC/PTX was assumed on the basis of previous works (*Baabur-Cohen et al., 2017*).

## PTX release from dPG conjugates

The release of PTX from the conjugates was analyzed by HPLC as previously described (*Sousa-Herves et al., 2015*). Briefly, the release of PTX in human plasma, pH 7.4, pH 5.0, and pH 2.0 was determined using a Knauer Smartline-HPLC system with an internal UV absorption detector (λ = 227 nm) and EZIChrom software. A Hypersil ODS C18 column (Thermo Fischer Scientific, MA, USA; 100 mm × 4.6 mm, particle Size: 5 µm) with a direct-connect guard column C18 was employed. Acetonitrile–water (65:35) was used as the mobile phase at a flow rate of 1.0 mL min$^{-1}$ under isocratic regime. The injection volume was 20 µL and each injection was performed in triplicate. Stock solutions of PTX-Bz in acetonitrile were prepared and assessed by reverse phase HPLC (RP-HPLC) in order to obtain a calibration curve for PTX-Bz (0.5–5 µg, R = 0.999) (Retention time: 2.9 min). The release profile from dPGS-PTX and dPG-PTX at different pHs was analyzed. For that purpose, the conjugates (constant PTX concentration) were incubated with universal Britton-Robinson buffer (BRB) of pH 7.4, 5.0, and 2.0 and human plasma (1:3 diluted with PBS). Samples were maintained at 37°C under continuous shaking, and aliquots (100 µL) were taken at different time intervals (1, 3, 5, 7, and 24 hr). The aqueous aliquots were mixed with 1 mL of Et$_2$O-CHCl$_3$ (1:1), vortexed for 2 min, and the phases were separated by centrifugation (10 min, 10,000 rpm, rt). The organic phase (900 µL) were taken for each sample, concentrated under vacuum, reconstituted with 200 µL of acetonitrile and analyzed by RP-HPLC. As control experiments, free PTX-Bz was incubated at the same concentration, extracted under identical circumstances and then analyzed by RP-HPLC.

## Dynamic Light Scattering (DLS) and Zeta potential determination

Measurements of mean hydrodynamic diameter and zeta-potential of the non-targeted dendritic conjugate (dPG-PTX) and the targeted dendritic conjugate (dPGS-PTX) were performed using a ZetaSizer Nano ZS instrument with an integrated 4 mW He-Ne laser (λ = 633 nm; Malvern Instruments Ltd., Malvern, Worcestershire, UK). Samples were prepared by dissolving 1 mg conjugate in 1 mL PBS 15 mM (pH 7.4). All measurements were performed at 25°C using polystyrol/polystyrene (10 × 4 × 45 mm) cells for DLS analysis and folded capillary cells (DTS 1070) for zeta-potential measurements. Results are representative of 3 repeats.

### Scanning electron microscope (SEM)

Samples were prepared by dissolving 0.1 mg dendritic conjugate in 1 mL DDW. Samples were dropped on a silicon wafer and blotted with cellulose paper. The dry samples were coated with 4 nm layer of Cr. SEM images were taken using Quanta 200 FEG Environmental SEM (FEI, Oregon, USA) at high vacuum and 5.0 KV. Diameters were measured by measureIT software.

### Cell culture

U-87 MG and U251 human glioblastoma cell lines were purchased from the American Type Culture Collection (ATCC, Manassas, VA, USA) and grown in DMEM supplemented with 10% FBS, 100 U/mL Penicillin, 100 µg/mL Streptomycin, and 2 mM L-glutamine. GL261 cells were obtained from the National Cancer Institute (Frederick, MD, USA) and grown in DMEM supplemented with 10% FBS, 100 U/mL Penicillin, 100 µg/mL Streptomycin, and 2 mM L-glutamine. U-87 MG and GL261 cells

were labeled with mCherry as previously described (Satchi-Fainaro et al., 2012). Human umbilical vein endothelial cells (HUVEC) were purchased from Lonza, Switzerland and cultured in EGM-2 medium (Lonza, Switzerland). Human astrocytes were purchased from ScienCell and cultured in astrocytes medium (ScienCell, CA, USA). Human cerebral microvascular endothelial cells (hCMEC/D3) were purchased from Merck and cultured in EndoGRO MV complete medium (Merck, Germany). Cells were routinely tested for mycoplasma contamination with a mycoplasma detection kit (Biological Industries, Israel). All cells were grown at 37°C in 5% $CO_2$.

## Human primary glioblastoma cells

Fresh human glioblastoma tissues were obtained from Tel Aviv Medical Center (Tel-Aviv, Israel) in accordance with a protocol approved by the IRB committee. Tumor tissues were obtained during surgical resection, kept in cold PBS and processed within 40 min. In order to isolate tumor cells and generate cells monolayer, tumor specimens were dissected to 0.5 mm pieces, plated in 6 cm plates and cultured with 1 mL DMEM supplemented with 10% FBS, 100 U/mL Penicillin, 100 μg/mL Streptomycin and 2 mM L-glutamine. Following continuous media replacement, viable cancer cells remained attached to culture plates and kept growing in culture, while stroma and cell debris were washed. Cells were routinely tested for mycoplasma contamination with a mycoplasma detection kit (Biological Industries, Israel). All cells were grown at 37°C in 5% $CO_2$.

## Bioinformatics analysis of P-selectin expression

SELP gene expression in brain tissues was obtained from three studies: healthy brain tissue from GTEx data (http://www.nature.com/ng/journal/v45/n6/abs/ng.2653.html) containing 1259 samples, Lower grade Glioma and Glioblastoma from TCGA obtained from cBioPortal (https://www.ncbi.nlm.nih.gov/pmc/articles/PMC4160307/) containing 530 and 578 samples, respectively. We used Transcripts per kilobase per million (TPM) levels to compare across these studies (obtained through R 'sweep' function) and compared the resulting distributions with Wilcoxon rank-sum test. Kaplan-Meier survival curves were obtained from TCGA data of glioblastoma patients with high or low P-selectin expression (using 63 samples with top and bottom 10% of SELP expression). SELP expression in various cancer types was obtained from the TCGA data portal. A graph was generated using the cBioPortal for cancer genomics and sorted by the median. SELP gene expression in healthy and cancerous tissues was obtained from XENA database (http://xena.ucsc.edu/public-hubs/), which normalizes gene expression from TCGA and GTEx together. For each tissue type, the distribution of SELP gene in healthy versus cancerous tissue was compared via a Wilcoxon one-sided rank sum test, and the boxplots are presenting the distribution of SELP in each tissue.

## Confocal microscopy

For intracellular trafficking analysis of the dendritic conjugates, U-87 MG cells were plated on 13 mm cover glass ($1 \times 10^5$ cells/cover glass) and were allowed to form a monolayer for 24 hr. Cells were then added with dye-labeled dPGS-PTX-IDCC or dPG-PTX-IDCC (0.5 μM equivalent PTX). Five minutes prior to each time point, cells were incubated with LysoTracker Red DND-99 (Thermo Fisher Scientific, MA, USA) and then washed and fixed using 4% paraformaldehyde (PFA) for 20 min. Cells were then stained with anti α-tubulin antibody (BioLegend, CA, USA) for 2 hr, followed by incubation with FITC-labeled secondary antibody (Jackson ImmunoResearch Laboratories Inc., PA, USA) for 1 hr. Cover glasses were then mounted by ProLong Gold mountant with DAPI (Thermo Fischer Scientific, MA, USA). Cellular uptake and co-localization between the polymer and lysosome were monitored with a Leica TCS SP5 confocal imaging system (Leica Microsystems, Wetzlar, Germany).

For in vivo intracranial tumor targeting analysis of the dendritic conjugates, 5 μm thick brain sections were mounted with ProLong Gold antifade mountant with DAPI (Thermo Fischer Scientific). Fluorescent signals of mCherry-labeled tumor cells, IDCC-labeled dendritic conjugates and DAPI were imaged using Leica TCS SP8 confocal imaging system (Leica Microsystems, Wetzlar, Germany).

## Flow cytometry

For cellular uptake of IDCC-labeled dendritic conjugates, patient-derived human glioblastoma cells (hGB1) were plated in 6-well plates ($2.5 \times 10^5$ cells/well) and allowed to form a monolayer for 24 hr. Cells were then added with dye-labeled dPGS-PTX-IDCC or dPG-PTX-IDCC (0.5 μg/ml equivalent

IDCC). At each time point, cells were harvested with trypsin, washed with PBS and analyzed for IDCC fluorescent intensity using Attune NxT Acoustic Focusing Flow Cytometer (Thermo Fisher Scientific, MA, USA). To evaluate the internalization mechanism, cells were incubated with 0.1, 1 and 10 μM P-selectin inhibitor (Tocris Bioscience, UK) for 1 hr prior to treatment with the IDCC-labeled conjugates. Cellular uptake of the conjugates was then evaluated as abovementioned.

For analysis of P-selectin expression, GL261, U-87 MG and patient-derived human glioblastoma cell lines (hGB) were harvested with trypsin and immediately washed with serum containing media followed by PBS supplemented with 2% BSA and 0.1% sodium azide. GL261 cells were incubated with FITC-labeled rat anti-mouse P-selectin antibody (BD Bioscience, USA) for 1.5 hr. U-87 MG and hGBM cells were incubated with anti- human P-selectin antibody (R&D Biosystems, MN, USA) for 2 hr, washed and incubated for 1 hr with either FITC-labeled or TRITC-labeled secondary antibody (Jackson ImmunoResearch Laboratories Inc., PA, USA). Fluorescent intensity was analyzed using either FACSAria flow cytometer (BD Biosciences, USA) or Gallios flow cytometer (Beckman Coulter, CA, USA).

For apoptosis analysis, HUVEC were treated with 1 ng/ml TSP-1 PM, 10 nM PTX and their combination for 72 hr. Cells were then harvested with Trypsin, washed with PBS and incubated with FITC-labeled annexin V and PI (MEBCYTO Apoptosis Kit, MBL International, UK), according to the manufacturer's protocol. Fluorescent intensity was analyzed using Attune NxT Acoustic Focusing Flow Cytometer (Thermo Fisher Scientific, MA, USA).

## Cell cycle analysis

Sixty percent confluent U-87 MG cells were treated with free PTX, dPGS-PTX or dPG-PTX (equivalent PTX) for 2, 4, 8, and 24 hr. At each of these time points cells were harvested and fixed in 70% ethanol. The cells were then washed with PBS and resuspended in the presence of RNase (1 μg/mL) and propidium iodide (PI, 50 μg/mL) for 30 min. Cell cycle histograms were generated using a BD Accuri C6 flow cytometer (BD Biosciences, USA).

## Cell viability assay

Human glioblastoma cell lines U-87 MG and U251 (10,000 cells/well) were plated onto 24-well culture plates in DMEM supplemented with 2% FBS and incubated for 24 hr. HUVEC (10,000 cells/well) were plated onto 24-well culture plates in EBM-2 supplemented with 5% FBS and incubated for 24 hr. The medium was then replaced with DMEM supplemented with 10% FBS or EGM-2. Cells were then exposed to PTX, dPGS-PTX or dPG-PTX at serial dilutions, at equivalent dose of free PTX. Number of viable cells was counted by a Z1 Coulter Counter (Beckman Coulter) following 72 hr of incubation.

## Tumor spheroids

Multicellular tumor spheroids were prepared using the hanging-drop method, in which drops of cells suspension are held hanging from the bottom of an inverted tissue-culture plate until cells agglomerate spontaneously at the lower part of the drop due to gravity (*Timmins and Nielsen, 2007*). Here, 3D tumor spheroids were formed from a mixture of multiple glioblastoma cell populations to better simulate the in vivo characteristics of glioblastoma in vitro. Briefly, cells suspension of human astrocytes, mCherry-labeled patient-derived glioblastoma cells and GFP-labeled HUVEC or hCMEC/D3 cells (80,000 cells/mL; 1:1:2 ratio) was prepared in endothelial growth medium (EMG)−2 supplemented with 0.24 w/v% methyl cellulose. Cells were deposited in 25 μL droplets on the inner side of a 20 mm dish and incubated for 48 hr at 37°C when the plate is facing upside down to allow for spheroid formation. Spheroids were then embedded in matrigel, seeded in a 96-well plate and treated with 1 ng/mL TSP-1 PM, 10 nM PTX and their combination. 3D spheroid invasion was visualized following 48 hr using EVOS FL Auto cell imaging system (Thermo Fisher Scientific).

## Aortic ring assay

Aortas were resected from a Balb/c mice, sliced to 1 mm pieces and placed in 48 wells plate coated with Matrigel basement membrane (250 μL/well; 10 mg/mL) on ice following 30 min incubation at 37°C. Additional Matrigel basement membrane (250 μL/well; 10 mg/mL) was added and allowed to polymerize at 37°C for 30 min. Conditioned media from U-87 MG cells, alone or with 1 ng/mL TSP-1

PM, was added (300 µL). Sprouting of endothelial cells from the aorta was imaged following 8 days incubation at 37°C using Nikon TE2000E inverted microscope integrated with Nikon DS5 cooled CCD camera by 15X objective, brightfield illumination.

## Miles vascular permeability assay

Balb/c male mice were injected intraperitoneally with 100 mg/kg TSP-1 PM or 5% dextrose daily for two days. Then, Evans Blue dye (100 mL of a 1% solution in 0.9% NaCl) was injected into the retro-orbital plexus of the mice. Ten minutes later, 50 µL of human VEGF (1 ng/mL) or PBS were injected intradermally into the pre-shaved back skin. Twenty minutes later, the animals were killed, and an area of skin that included the entire injection site was removed. Evans Blue dye was extracted from the skin by incubation in formamide for 5 days at room temperature. Absorbance of the extracted dye was measured at 620 nm. Data is expressed as mean ± standard error of the mean (s.e.m.).

## Animal studies

In order to evaluate targeting efficacy of the dendritic conjugates, mCherry-labeled U-87 MG cells (2 × 10$^5$) were stereotactically implanted into the striatum of 6 weeks old, male, SCID mice (Envigo CRS, Israel) and mCherry-labeled GL261 cells (2 × 10$^5$) were stereotactically implanted into the striatum of 6 weeks old C57BL mice (Envigo CRS, Israel). Once the tumor was visualized by intravital fluorescence imaging system (CRI Maestro, MA, USA), mice were administered with dye-labeled dPG-PTX-IDCC or dPGS-PTX-IDCC (50 µM equivalent IDCC) and monitored overtime for IDCC fluorescent signal. At the end of each time period, mice were euthanized and immediately perfused with PBS followed by 4% formaldehyde. Mice brains were then harvested and embedded in an optimal cutting temperature (OCT) compound followed by frozen-sectioning.

For drug efficacy studies, GL261 murine glioblastoma cells (2 × 10$^5$) were stereotactically implanted into the striatum of 5–6 weeks old male C57BL/6 mice (Envigo CRS, Israel). One week following tumor cells implantation, mice were administered IP every other day with dPGS, dPGS-PTX, dPG, dPG-PTX or saline for 2 weeks. dPGS-PTX and dPG-PTX were administered at 15 mg/kg equivalent PTX dose. dPGS and dPG were administered at a dose equivalent to the drug-conjugated formulation. Hundred mg/kg of TSP-1 were administered IP every day for 2 weeks. Mice administered with oral TMZ (100 mg/kg; every day for 5 days) were used as a standard treatment control group. Mice survival was monitored daily and blood was extracted 45 days following treatment initiation for blood count and biochemistry analysis. An additional efficacy study was performed with mCherry-labeled U-87 MG cells. Cells were (2 × 10$^5$) were stereotactically implanted into the striatum of 6–8 weeks old male SCID mice (Envigo CRS, Israel). Tumor growth was monitored using an intravital fluorescence imaging system (CRI Maestro) twice a week. Mice were randomized into groups according to the fluorescent signal detected in the brain and administered systemically with the treatments for 7 days. Mice were then euthanized and immediately perfused with PBS followed by 4% formaldehyde. Mice brains were then harvested and embedded in OCT followed by frozen-sectioning.

## Immunohistochemistry

Formalin-fixed, paraffin-embedded samples of tumor nodules were cut into 5 µm thick sections. Paraffin sections were deparaffinized, rehydrated, and stained by hematoxylin and eosin (H&E). For immunohistochemistry staining, slides were deparaffinized and pre-treated with 10 mM citrate, pH 6.0 for 50 min in a steam pressure cooker (BioCare Medical, Walnut Creek, CA). All further steps were performed at rt in a hydrated chamber. Slides were covered with Peroxidase Block (Merck, Germany) for 5 min to quench endogenous peroxidase activity, followed by incubation with 10% of goat serum in 50 mM Tris-HCl, pH 7.4, for 30 min to block nonspecific binding sites. Primary mouse anti-P-selectin (R&D) rabbit anti-VEGF (Santa Cruz Biotechnology, Inc., CA, USA) or mouse anti-α-SMA (Sigma Aldrich) were applied in 1% goat serum in Tris-HCl, pH 7.4 at RT for 1 hr. A broad spectrum biotinylated secondary antibody was added for 1 hr. Slides were then incubation with streptavidin-horseradish peroxidase conjugate for 30 min (Histostain, Life Technologies, CA, USA). Following further washing, immunoperoxidase staining was developed using ImmPACT DAB diluent kit (Vector Laboratories, CA, USA) per the manufacturer instructions and counterstained with hematoxylin.

For immunostaining of intracranial brain tumors, brains embedded in OCT were cryosectioned into 5 µm thick sections. Immunostaining was performed using the BOND RX automated IHC stainer

(Leica Biosystems). Briefly, slides were incubated with goat serum (10% goat serum in PBSX1 + 0.02% Tween-20 + 0.02% Gelatin) for 30 min to block non-specific binding sites. Slides were then added with mouse anti-human P-selectin (R&D, 1:20 dilution), rat anti-mouse CD31 (BD Bioscience; 1:20 dilution) rabbit anti-Fas antibody (Abcam, MA, USA; 1:25 dilution) or rabbit anti-Fas ligand antibody (Abcam; 1:100 dilution). Following 1 hr incubation, slides were incubated with the following secondary antibodies for an additional 1 hr: goat anti-mouse Alexa-488 (Jackson Immunoresearch, 1:400 dilution) for P-selectin; goat anti-rat Alexa-488 (Jackson Immunoresearch; 1:350 dilution) for CD31; and goat anti-rabbit Alexa-488 (Abcam; 1:150 dilution) for Fas and Fas-L. Sections were then mounted with ProLong Gold antifade mountant with DAPI (Thermo Fischer Scientific) and imaged using EVOS FL Auto cell imaging system (Thermo Fisher Scientific).

## Motor coordination test

PTX-related motor-coordination was assessed using a Rotarod apparatus (Columbus Instruments, OH, USA). Animals were acclimated to the protocol during the 3 days before each of the testing dates. Initial speed was 1.6 rpm, with acceleration rate of 4 rpm per minute. Animals were tested three times during each session with at least 2 min of rest between each test. The best performance for each testing date (before and after treatment) was recorded.

## Statistical methods

Data are expressed as mean ± standard deviation (s.d.) for in vitro assays or ± standard error of the mean (s.e.m.) for in vivo assays. Statistical analysis for two sets of data was performed using an unpaired t-test. Statistical significance of differences in overall survival was determined using log-rank test. The distribution of SELP gene in healthy *versus* cancerous tissues was compared via a Wilcoxon one-sided rank sum test. Significance was defined as $p < 0.05$.

## Acknowledgements

RS-F and RH would like to thank the German-Israel Foundation (GIF) for financial support. RS-F, MC and RH thank Tel Aviv University and Freie Universität Berlin for financial support to organize collaborative workshops. The Satchi-Fainaro laboratory's research leading to these results has received partial funding from the European Research Council under the European Union's Seventh Framework Programme (FP/2007–2013)/ERC Consolidator Grant Agreement n. [617445], from the Israel Science Foundation (918/14), from Nancy and Peter Brown friends of the Israel Cancer Association ICA USA, in Memory of Kenny and Michael Adler, and from the Morris Kahn Foundation. RH acknowledges financial support from the Bundesmiminsterium für Bildung und Forschung (BMBF) within the Biotransporter project (project number: 13N11536) and the SFB 765. MC gratefully acknowledges financial support from the BMBF through the NanoMatFutur award (13N12561) and the focus area Nanoscale of the Freie Universität Berlin (http://www.nanoscale.fu-berlin.de). ASH is grateful for a Marie Curie Intra-European Fellowship (302717). We thank Prof. Adit Ben-Baruch and Yulia Liubomirski for their professional assistance with the tumor spheroids model. We thank Lior Bikovski for his assistance with the behavioral studies performed at the Myers neuro-behavioral core facility at the Sackler Faculty of Medicine, Tel Aviv University.

## Additional information

### Funding

| Funder | Grant reference number | Author |
| --- | --- | --- |
| European Research Council | 617445 | Ronit Satchi-Fainaro |
| Israel Science Foundation | 918/14 | Ronit Satchi-Fainaro |
| Israel Cancer Association | 20150909 | Ronit Satchi-Fainaro |
| Bundesministerium für Bildung und Forschung | 13N11536 | Rainer Haag |
| Bundesministerium für Bildung und Forschung | 13N12561 | Marcelo Calderón |

The funders had no role in study design, data collection and interpretation, or the decision to submit the work for publication.

## Author contributions
Shiran Ferber, Galia Tiram, Ana Sousa-Herves, Conceptualization, Data curation, Formal analysis, Validation, Investigation, Visualization, Methodology, Writing—original draft, Writing—review and editing; Anat Eldar-Boock, Anna Scomparin, Data curation, Validation, Investigation, Visualization, Methodology, Writing—review and editing; Adva Krivitsky, Laura Isabel Vossen, Data curation, Formal analysis, Validation, Investigation, Visualization, Methodology, Writing—review and editing; Eilam Yeini, Kai Licha, Validation, Investigation, Visualization, Methodology, Writing—review and editing; Paula Ofek, Data curation, Formal analysis, Validation, Investigation, Visualization, Methodology; Dikla Ben-Shushan, Data curation, Investigation, Visualization, Methodology; Rachel Grossman, Zvi Ram, Resources, Writing—review and editing; Jack Henkin, Investigation, Methodology, Writing—original draft, Writing—review and editing; Eytan Ruppin, Data curation, Software, Formal analysis, Investigation, Methodology, Writing—review and editing; Noam Auslander, Data curation, Formal analysis, Investigation, Methodology; Rainer Haag, Marcelo Calderón, Conceptualization, Resources, Supervision, Funding acquisition, Investigation, Methodology, Project administration, Writing—review and editing; Ronit Satchi-Fainaro, Conceptualization, Resources, Formal analysis, Supervision, Funding acquisition, Investigation, Methodology, Writing—original draft, Project administration, Writing—review and editing

## Author ORCIDs
Galia Tiram [ID] http://orcid.org/0000-0002-5425-8088
Marcelo Calderón [ID] http://orcid.org/0000-0002-2734-9742
Ronit Satchi-Fainaro [ID] http://orcid.org/0000-0002-7360-7837

## Ethics
Human subjects: Experiments involving human tissues were performed with the approval of the Institutional Review Board (IRB) and in compliance with all legal and ethical considerations for human subject research (approval no. 0735-13-TLV). Single human plasma was obtained from a healthy consented unmedicated donor according to German ethical guidelines.

Animal experimentation: All animals were housed in the Tel Aviv University animal facility. The experiments were approved by the animal care and use committee (IACUC) of Tel Aviv University (approval no. 01-12-064, 01-12-065) and conducted in accordance with NIH guidelines.

## Decision letter and Author response
Decision letter https://doi.org/10.7554/eLife.25281.028
Author response https://doi.org/10.7554/eLife.25281.029

## Additional files
### Supplementary files
• Transparent reporting form
DOI: https://doi.org/10.7554/eLife.25281.025

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
