## [Decision Letter]

Thank you for submitting your article "Co-targeting the tumor endothelium and P-selectin-expressing glioblastoma cells leads to complete tumor eradication" for consideration by *eLife*. Your article has been favorably reviewed by two peer reviewers, and the evaluation has been overseen by Charles Sawyers as the Senior Editor and Reviewing Editor. The reviewers have opted to remain anonymous.

The reviewers have discussed the reviews with one another and the Reviewing Editor has drafted this letter to crystallize our concerns going forward. We feel the work is important and interesting but key issues remain unresolved that must be addressed satisfactorily to produce an acceptable manuscript.

Summary:

The manuscript "Co-targeting the tumor endothelium and P-selectin-expressing glioblastoma cells leads to complete tumor eradication" is the study of a drug delivery strategy employing a P-selectin-targeted dendrimer-based nanomaterial for the localization of therapy to tumors in an orthotopic glioblastoma model. The manuscript details targeting and efficacy studies, as well as a synergistic combination of P-selectin-targeted paclitaxel with an angiogenesis inhibiting peptide.

Essential revisions:

The revised manuscript must address the following three points:

1) The mechanism for synergy with ABT-898.

The mechanism of the drug synergy was not explored, except via discussion, where the author deemed it "out of the scope" of the manuscript. Given that the combination of dPGS-PTX and the TSP-1 peptide confers a much enhanced efficacy as compared to dPGS-PTX alone, this therapeutic result, which is the most striking in the paper, demands some degree of mechanistic understanding. Some experimentation should be provided to rule in or rule out at least one or two of the mechanistic possibilities.

The drug combination of dPGS-PTX with ABT-898, while interesting and apparently effective, is only examined at a very descriptive level and many questions remain why this combination was chosen and how the combination results in greater antitumor activity at a molecular and histopathological level.

2) Provide more data on where dPGS-PTX is acting (tumor vs. vasculature and the role of P-selectin expression in tumor cells).

A conspicuous absence is an understanding of where the dPGS-PTX localizes and acts in and around the tumor with respect to vasculature and/or tumor cells. The author noted a relatively limited infiltration of the construct within the tumor. Immunofluorescence staining of P-selectin and/or vasculature within the tumors harboring the fluorescent dPGS-PTX-IDCC may impart some understanding of whether the nanocarriers localize in at the vasculature and whether P-selectin expression in that vasculature may aid localization and/or penetration of the nanocarriers through the vasculature. Staining for apoptosis in the brain tissue may aid in the understanding of whether the efficacy of the nanocarrier is due to action on vasculature or tumor cells.

One of the surprising findings of the study is that glioma cells apparently express/overexpress P-selectin. This could provide rationale to further pursue dendritic polyglycerol sulfate (dPGS) as particularly relevant drug delivery strategy in glioma/GBM, but should be corroborated by additional evaluation of P-selectin expression in a larger panel of glioma cell lines, other cancer cell lines, and human glioma tumor samples compared to other cancer types (TCGA datasets). The authors should also modulate P-selectin expression levels in tumor cells (knockdown/knockout/overexpression) and document how P-selectin levels affect the in-vivo response to dPGS-PTX and dPG-PTX. At least in-vitro, both PGS-PTX and dPG-PTX conjugates appear to induce cell cycle arrest and cell death to a similar degree, even though the kinetics of internalization into tumor cells appear to be vastly different.

3) Show evidence of benefit in another model.

The in-vivo findings appear to be based on experiments with a single experimental model (C57BL/6 mice inoculated intracranially with murine GL261 glioma cells) and treatment started within seven days of injection of the tumor cells. Documentation of tumor formation in all mice prior to treatment, examination of tumors/brains at the end of the treatment period, and inclusion of additional, patient-derived GBM tumor sphere xenografts models would strengthen the current conclusions.

[Editors' note: further revisions were requested prior to acceptance, as described below.]

Thank you for resubmitting your work entitled "Co-targeting the tumor endothelium and P-selectin-expressing glioblastoma cells leads to a remarkable therapeutic outcome" for further consideration at *eLife*. Your revised article has been favorably evaluated by Charles Sawyers as the Senior Editor and Reviewing Editor, and two reviewers.

The manuscript has been improved but there are a few remaining issues that need to be addressed before acceptance, as outlined below:

The authors have generated new data to address several key concerns. Specifically, they have: (1) provided in-vivo and in-vitro data in a second experimental GBM model (U87 human GBM cells), (2) identified several potential explanations for the synergy between PTX and TSP-1 PM (ABT-898), and (3) conducted experiments with a P-selectin inhibitor to strengthen their previous data regarding drug specificity. The new data considerably strengthens the manuscript. But the remaining question whether tumor cells or endothelial cells/blood vessels are the primary target of the PTX/TSP-1 PM combination is not completely resolved by the immunofluorescence experiment staining of GL261 and U87 tumors with dPGS-PTX-I DCC showing more consistent colocalization with CD31.

We suggest the authors make the following modifications to the text:

1) Modify the Discussion to acknowledge that the precise cellular target of P-selectin-targeted dPDGS-PTX (tumor versus vasculature) is not known and would likely require experiments in P-selectin knockout tumor cells.

2) Comment on the decision to focus on GBM given that GBMs express lower levels of P-selectin than just about any other human cancer (Figure 3—figure supplement 1).

3) Provide further experimental details and/or literature references in the Materials and methods section on the tumor spheroid model, to avoid confusion with the more widely used models (e.g., tumor sphere culture, IPS-derived organoids)

4) Provide further details regarding the P-selectin inhibitor (beyond the chemistry publication introducing a series of compounds).

---

## [Author Response]

Essential revisions:The revised manuscript must address the following three points:1) The mechanism for synergy with ABT-898.The mechanism of the drug synergy was not explored, except via discussion, where the author deemed it "out of the scope" of the manuscript. Given that the combination of dPGS-PTX and the TSP-1 peptide confers a much enhanced efficacy as compared to dPGS-PTX alone, this therapeutic result, which is the most striking in the paper, demands some degree of mechanistic understanding. Some experimentation should be provided to rule in or rule out at least one or two of the mechanistic possibilities.The drug combination of dPGS-PTX with ABT-898, while interesting and apparently effective, is only examined at a very descriptive level and many questions remain why this combination was chosen and how the combination results in greater antitumor activity at a molecular and histopathological level.

First, we addressed this concern by determining the synergistic effect of PTX and TSP-1 PM (ABT-898) in vitro on cells proliferation and invasion and deciphering the mechanism responsible for this synergism. We demonstrate that combination treatment enhanced apoptosis of endothelial cells and inhibited sprouting of patient-derived glioblastoma spheroids to a greater extent compared to each treatment alone.

The following text was added to the Results section:

“To evaluate the therapeutic efficacy of TSP-1 PM and PTX combination, we performed an annexin V/propidium iodide (PI) apoptosis assay on HUVEC. […] Combination of TSP-1 PM and PTX inhibited both endothelial and glioblastoma cells' sprouting to a greater extent compared to the other treatment groups (Figure 7).”

Second, we hypothesized that TSP-1 PM can improve the efficacy of dPGS-PTX by “normalizing" local vasculature to allow improved delivery of chemotherapy. We show that TSP-1 PM stabilizes blood vessels in GL261 glioblastoma inoculated intracranially in C57Bl/6J mice, thereby enhancing the internalization of the dendritic conjugates.

The following text was added to the Results section:

“It has been demonstrated previously that TSP-1 PM can increase the concentration of chemotherapeutic agents at the tumor site through vessel normalization (Yap et al., 2005). […] αSMA staining confirmed that TSP-1 PM stabilizes blood vessels in GL261 tumors (Figure 7), thereby enhancing the internalization of the dendritic conjugates, as demonstrated by confocal imaging of tumor sections (Figure 7).”

In addition, we hypothesized that the synergistic effect is due to enhanced apoptosis via Fas and Fas Ligand. Immunostaining of U-87 MG tumors for Fas and Fas-L demonstrated that the combination treatment induces the expression of these two apoptotic markers.

The following text was added to the Results section:

“To understand by which mechanism dPGS-PTX and TSP-1 PM synergize in vivo, the effect of the treatments on the Fas/Fas-ligand (Fas-L) apoptosis pathway was analyzed immunohistochemically. […] Furthermore, decreased CD31 staining was demonstrated in tumors treated with our dendritic conjugates (Figure 8).”

2) Provide more data on where dPGS-PTX is acting (tumor vs. vasculature and the role of P-selectin expression in tumor cells).A conspicuous absence is an understanding of where the dPGS-PTX localizes and acts in and around the tumor with respect to vasculature and/or tumor cells. The author noted a relatively limited infiltration of the construct within the tumor. Immunofluorescence staining of P-selectin and/or vasculature within the tumors harboring the fluorescent dPGS-PTX-IDCC may impart some understanding of whether the nanocarriers localize in at the vasculature and whether P-selectin expression in that vasculature may aid localization and/or penetration of the nanocarriers through the vasculature. Staining for apoptosis in the brain tissue may aid in the understanding of whether the efficacy of the nanocarrier is due to action on vasculature or tumor cells.

To address this interesting point, we stained sections of mCherry-labeled U-87 MG and GL261 tumors treated with dPGS-PTX-IDCC with CD31 and P-selectin.

We added the following text to the Results section:

“To understand where dPGS-PTX-IDCC acts within the tumor, we stained U-87 MG and GL261 tumor sections for P-selectin and CD31. […] This suggests that P-selectin-targeted dPGS-PTX works primarily on P-selectin-expressing tumor vasculature and only then penetrates the tumor and affects P-selectin-expressing glioblastoma cells.”

One of the surprising findings of the study is that glioma cells apparently express/overexpress P-selectin. This could provide rationale to further pursue dendritic polyglycerol sulfate (dPGS) as particularly relevant drug delivery strategy in glioma/GBM, but should be corroborated by additional evaluation of P-selectin expression in a larger panel of glioma cell lines, other cancer cell lines, and human glioma tumor samples compared to other cancer types (TCGA datasets).

In order to address the reviewers' questions regarding the "additional evaluation of P-selectin expression in a larger panel of glioma cell lines, other cancer cell lines, and human glioma tumor samples compared to other cancer types (TCGA datasets)", we first analyzed by FACS the expression of P-selectin on several glioblastomas (5 in the upper line including human, murine, and 3 patient-derived cells); melanoma cells (5 cell lines in the second line), and mammary adenocarcinoma (4TI, DA3), PDAC (KPC), and osteosarcoma (K7M2) (see Author response image 1). As can be seen, all but one (K7M2 murine osteosarcoma) had high expression of P-selectin.

Furthermore, we have been collaborating lately with Eytan Ruppin from the University of Maryland (https://www.umiacs.umd.edu/people/eruppin) to address the question regarding P-selectin expression in TCGA datasets. We have added him and Noam Auslander to the authors' list of this current paper. A TCGA and Genotype-Tissue Expression (GTEx) data search demonstrated that P-selectin is highly expressed in glioblastoma tumors, compared to its expression in the normal brain. Interestingly, when we were looking into the role of P-selectin in glioblastoma, we found that its expression correlated with survival of glioblastoma patients as seen in the Kaplan-Meier curves obtained from cBioPortal.

The following text was added to the Results section:

“To validate these findings, we looked at The Cancer Genome Atlas (TCGA) data analyzing the expression of P-selectin (SELP) in glioblastoma as well as other cancer types. […] A gene-expression-based survival analysis using TCGA data obtained from cBioPortal showed that P-selectin expression correlated with survival of glioblastoma patients (Figure 3).”

The authors should also modulate P-selectin expression levels in tumor cells (knockdown/knockout/overexpression) and document how P-selectin levels affect the in-vivo response to dPGS-PTX and dPG-PTX. At least in-vitro, both PGS-PTX and dPG-PTX conjugates appear to induce cell cycle arrest and cell death to a similar degree, even though the kinetics of internalization into tumor cells appear to be vastly different.

This is an interesting point commonly discussed in the field of drug delivery. It is well known that differences in the activity of targeted versus not-targeted nanomedicines (P-selectin-targeted vs non-targeted polymer-drug conjugates in our case) cannot be demonstrated in vitro as in a standard cytotoxicity assay, the cells are bathed for 72 hours with medium containing both compounds. Within 72 hours, most nanomedicines will internalize via endocytosis, whether it is fluid-phase pinocytosis (for the non-targeted conjugate) or receptor-mediated endocytosis (for the P-selectin-targeted conjugate). To that end, we ran a pulse and chase assay where we treated U-87 MG glioblastoma cells with free PTX (which diffuses the fastest into the cells), with dPGS-PTX (which internalizes rapidly by receptor-mediated endocytosis) or by dPG-PTX (which passively internalizes via fluid-phase pinocytosis) and washed the wells after 30 minutes, leaving the cells for 72 hours and reading their viability thereafter. It can be seen that, as expected from these three compounds, there was a difference in IC_50_ exhibiting the lowest IC_50_ for free PTX (100 nM), intermediate for dPGS-PTX (400 nM) and the highest for the non-targeted dPG-PTX (N/A) (Figure 4—figure supplement 1). This explains that even in vitro, we can detect differences in activity due to differences in internalization kinetics. The phenomenon is expected to be greatly enhance in real in vivo settings when the compounds are flowing in the bloodstream and extravasating through the tumor leaky vessels binding to cells expressing P-selectin (or not).

This data was added to the Results subsection “In vitro anti-tumor and anti-angiogenic effects of dPGS-PTX”.

As for the modulation of P-selectin, we used a P-selectin inhibitor to evaluate the differences in internalization of IDCC-labeled dendritic conjugates by FACS. First, we added new flow cytometry data demonstrating the different internalization kinetics of the conjugates (Figure 2). Then, we demonstrated that cellular uptake of dPGS-PTX was inhibited following treatment with a P-selectin inhibitor (Figure 2). These studies, in addition to the studies we showed in the paper using our synthesized control dPG-PTX which does not bind to P-selectin, give a more comprehensive picture showing the dependence of the targeted conjugat's enhanced activity on its selective and rapid internalization.

The following text was added to the Results section:

“In order to evaluate the cellular uptake of the dendritic conjugates into glioblastoma cells, patient-derived hGB1 cells were incubated with IDCC-labeled dPGS-PTX and dPG-PTX for different periods of time. […] This demonstrated that dPGS-PTX internalized cells via P-selectin.”

3) Show evidence of benefit in another model.The in-vivo findings appear to be based on experiments with a single experimental model (C57BL/6 mice inoculated intracranially with murine GL261 glioma cells) and treatment started within seven days of injection of the tumor cells. Documentation of tumor formation in all mice prior to treatment, examination of tumors/brains at the end of the treatment period, and inclusion of additional, patient-derived GBM tumor sphere xenografts models would strengthen the current conclusions.

We would like to note that the in vitro studies were performed on several glioblastoma cell lines: murine GL261, human U-87 MG (Figure 2, Figure 3), human U-251 (Figure 4), several patient-derived cells and tissues, including FFPEs from several patients (Figure 3) and HUVEC (Figure 4). As for in vivo studies: the body distribution and tumor accumulation was performed on human U-87 MG tumor-bearing mice, and the antitumor efficacy was performed on murine GL261. Therefore, we were encouraged by the fact that both in vitro and in vivo, the dPGS-PTX was active on several glioblastoma and stromal cells. As for documentation of tumor formation, we have vast experience with intracranial injection of GL261 glioblastoma cells into C57BL/6 mice. Tumor take for the GL261 murine glioblastoma model is approximately 90%.

Nevertheless, to address the request of the reviewers, we performed two additional in vivo studies with mCherry-labeled GL261 and U-87 MG glioblastoma cells. The in vivo study with GL261 cells was performed to demonstrate body distribution and tumor accumulation of IDCC-labeled conjugates in another tumor model, in addition to U-87 MG. The results supported the findings obtained from U-87 MG tumors, demonstrating that the targeted sulfated dendrimer preferably accumulates in intracranial tumors. This data was added to Figure 5, which now contains results of both tumor models.

An additional efficacy study with U-87 MG cells was performed, to show benefit of the combination treatment in another tumor model. Mice bearing intracranial U-87 MG-mCherry tumors were administered systemically with the different treatments and tumor growth was monitored using an intravital fluorescence imaging system (CRI Maestro). In contrast to the previous in vivo survival study with GL261 cells, this study was designed as an “endpoint” study, which enabled to evaluate both the efficacy and the mechanism of synergy of the combination treatment by IHC analysis. The results of this study demonstrate that combination treatment with TSP-1 PM and dPGS-PTX inhibits U-87 MG tumor growth in SCID mice. We show that this inhibition was mediated by induction of the Fas/Fas-L apoptosis pathway.

The following text was added to the Results section:

“Having shown improved survival of C57BL/6 mice bearing GL261 tumors following treatment with our combination therapy, we set to evaluate its therapeutic efficacy and mechanism of synergy in an additional in vivo tumor model. […] Furthermore, decreased CD31 staining was demonstrated in tumors treated with our dendritic conjugates (Figure 8).”

To strengthen our findings further, in addition to the different cell types that were used in the submitted study, we established a 3D tumor spheroid model composed of patient-derived glioblastoma cells, endothelial cells and astrocytes, which better imitates the in vivo settings of glioblastoma. This model was used to demonstrate the different internalization kinetics of the dendritic conjugates (Figure 2) and the in vitro anti-tumor effect of TSP-1 PM and PTX combination (Figure 7; full data can be found in the answer to question no. 1).

The following text was added to the Results section:

“To further show the advantage of dPGS-PTX compared to its non-targeted control, we treated patient-derived glioblastoma tumor spheroids with IDCC-labeled dendritic conjugates. Similar to the results received in 2D culture, dPGS-PTX internalized into the spheroids more rapidly and efficiently compared to dPG-PTX (Figure 2).”

[Editors' note: further revisions were requested prior to acceptance, as described below.]

[…] 1) Modify the Discussion to acknowledge that the precise cellular target of P-selectin-targeted dPDGS-PTX (tumor versus vasculature) is not known and would likely require experiments in P-selectin knockout tumor cells.

The following text was added to the Discussion section:

“Nevertheless, the precise cellular target of our P-selectin-targeted dPGS-PTX remains unclear and warrants further investigation. Future studies using P-selectin-knockout tumor and endothelial cells may resolve this issue and uncover which of these cellular compartments is targeted by dPGS-PTX.”

2) Comment on the decision to focus on GBM given that GBMs express lower levels of P-selectin than just about any other human cancer (Figure 3—figure supplement 1).

Glioblastoma remains one of the deadliest forms of cancer. Survival rates of glioblastoma patients did not improve drastically over the past decades, despite great investigative efforts. Therefore, there is an unmet need for efficacious targeted therapy for glioblastoma that we aimed to address. As one of the major obstacles in glioblastoma therapy is crossing the blood-brain-barrier, we exploited a dendritic polyglycerol (dPG)-based conjugate, which we have previously shown to efficiently extravasate through the compromised BBB. The sulfated dPG (dPGS) was primarily used to target P-selectin expressed on activated endothelial cells in the tumor microenvironment of highly angiogenic glioblastomas. Surprisingly, we revealed that glioblastoma cells express P-selectin as well. We therefore performed a TCGA data analysis of P-selectin expression in glioblastoma compared to other cancer types. While the expression of P-selectin was indeed the lowest across various cancer types, it should be noted that P-selectin was markedly and significantly increased in glioma and glioblastoma compared to healthy brain tissue. Therefore, P-selectin-targeted therapy in glioblastoma may potentially result in effective tumor targeting, while minimizing side effects. Despite having shown this data in a glioblastoma model, this therapeutic approach can be exploited for any type of cancer with high P-selectin expression, of particular interest are pancreatic cancer and ccRCC which seem like an attractive target (Figure 3—figure supplement 2).

The paragraph describing the results of TCGA data analysis was revised accordingly:

“These data suggest that even though P-selectin expression in glioblastoma is low compared to other tumor types, it represents a suitable target for glioblastoma therapy due to its potential for achieving effective tumor targeting, while minimizing side effects. […] Of particular interest were pancreatic cancer (PAAD) and clear cell renal cell carcinoma (ccRCC), which also seem like attractive targets for P-selectin-targeted therapies (Figure 3—figure supplement 2).”

3) Provide further experimental details and/or literature references in the Materials and methods section on the tumor spheroid model, to avoid confusion with the more widely used models (e.g., tumor sphere culture, IPS-derived organoids).

The following text was revised in the Materials and methods section:

**“**Multicellular tumor spheroids were prepared using the hanging-drop method, in which drops of cells suspension are held hanging from the bottom of an inverted tissue-culture plate until cells agglomerate spontaneously at the lower part of the drop due to gravity (Timmins and Nielsen, 2007). […] 3D spheroid invasion was visualized following 48 h using EVOS FL Auto cell imaging system (ThermoFisher Scientific).”

The following literature reference was also added:

Timmins, N. E., and Nielsen, L. K. (2007). Generation of multicellular tumor spheroids by the hanging-drop method. Methods Mol Med 140, 141-151.

4) Provide further details regarding the P-selectin inhibitor (beyond the chemistry publication introducing a series of compounds).

The following text was added to the Results section:

**“**To evaluate whether the internalization of dPGS-PTX is P-selectin-dependent, we used a low molecular weight P-selectin inhibitor (KF 38789, Tocris, UK). This compound has been previously shown to inhibit the binding of U937 lymphocytes to immobilized P-selectin immunoglobulin, with an IC_50_ value of 1.97 µM (Ohta et al., 2001).”

Also, the reference introducing a series of compounds was replaced with a reference that addresses the specific compound used in this study:

Ohta, S., Inujima, Y., Abe, M., Uosaki, Y., Sato, S., and Miki, I. (2001). Inhibition of P-selectin specific cell adhesion by a low molecular weight, non-carbohydrate compound, KF38789. Inflamm Res 50, 544-551.

Additional data on P-selectin inhibitor (from the product data sheet, Catalog No. 2748):

Molecular Formula: C_19_H_21_NO_5_S

Molecular Weight: 375.44 g/mol.

Chemical structure as displayed here https://www.tocris.com/products/kf-38789_2748